# Local Housing Strategy: Analysis of Importance and Implementation in Machico Municipality, Madeira

Raul Alves [1,*], Sérgio Lousada [2,3,4,5,6], José Cabezas [4,5,7] and José Manuel Naranjo Gómez [3,4,5,8]

1 Machico City Council (CMM), Largo do Município Machico, 9200-099 Machico, Portugal
2 Department of Civil Engineering and Geology (DECG), Faculty of Exact Sciences and Engineering (FCEE), University of Madeira (UMa), 9000-082 Funchal, Portugal; slousada@staff.uma.pt
3 CITUR—Madeira—Research Centre for Tourism Development and Innovation, 9000-082 Funchal, Portugal; valoriza@ipportalegre.pt
4 VALORIZA—Research Centre for Endogenous Resource Valorization, Polytechnic Institute of Portalegre (IPP), 9000-082 Portalegre, Portugal; turismo@mail.uma.pt
5 Research Group on Environment and Spatial Planning (MAOT), University of Extremadura, 06071 Badajoz, Spain
6 RISCO—Civil Engineering Department of University of Aveiro, 3810-193 Aveiro, Portugal
7 Science Faculty, University of Extremadura, 06006 Badajoz, Spain
8 School of Agricultural Engineering, University of Extremadura, 06007 Badajoz, Spain
* Correspondence: raul@cm-machico.pt; Tel.: +351-967-474-596

**Abstract:** This article presents a detailed analysis of the local housing strategies (ELH) in the Municipality of Machico, Madeira, examining the diagnosis of housing needs, proposed solutions, and projected goals for the future. In addition to addressing the rehabilitation of private and public housing and the increase in housing supply, the study also explores how ELH relate to the new generation of housing policies and the public support program for promoting housing solutions for vulnerable people. It is important to note that the implementation of ELH in Machico also has a positive impact on the local landscape and climate resilience, promoting the conservation of important forest areas for the production and management of natural resources.

**Keywords:** environmental preservation; forest conservation; housing needs; housing rehabilitation; landscape resilience; Machico; Madeira





## 1. Introduction

Urban development is a complex process influenced by various factors, including housing policies, socioeconomic dynamics, and environmental considerations. This paper aims to elucidate the interplay between housing policies, urban resilience, and sustainability, using Machico, Madeira, as a case study.

Housing is a fundamental right for all, yet many communities face significant challenges to ensure that everyone has access to adequate and affordable housing. The municipality of Machico, located on the island of Madeira, is no exception to this problem. Housing policy has been a growing concern for the municipality, and the implementation of local housing strategies (ELH) is seen as a solution to improving the housing situation in the region. The aim of this article is to analyze the importance and implementation of ELH in Machico by diagnosing housing needs, proposing, and evaluating solutions, and projecting objectives for the future. The strategy aims to address the housing challenges in Machico, with an emphasis on the rehabilitation of private housing.

It is worth noting that national-level housing policies play a crucial role in shaping the context within which local strategies are developed. The national housing strategy (ENH) and the "1° Direito" program are among the key initiatives at the national level that aim to enhance housing conditions. These policies focus on promoting access to adequate housing

for individuals living in inadequate housing situations and facing financial constraints to secure decent housing solutions.

Landscape is a fundamental element for the quality of life of communities and should be considered in the implementation of local housing strategies (ELH). Landscape can be understood as a complex, dynamic, and multifunctional system that can provide essential ecosystem services to society, such as climate regulation, erosion and pollution control, biodiversity protection, among others [1,2]. Landscape resilience is an important aspect to be considered in the implementation of ELH, as it allows the landscape to adapt to changes and disturbances, such as extreme weather events and natural disasters, while maintaining its functions and ecosystem services [3,4].

Integrating resilient landscape strategies can ensure that vital ecosystem services are maintained even in the face of environmental disturbances and changes. Local housing strategies (ELH) can include landscape measures such as promoting urban biodiversity, using sustainable building materials, and incorporating nature-based urban design. Various studies have highlighted the importance of resilient landscape measures in promoting sustainability and the resilience of the local landscape. Sheppard [5] emphasized the significance of integrating landscape planning into urban planning to increase cities' resilience to climate change. Sustainable building materials and nature-based urban design have also been identified as effective measures to enhance the resilience of cities and improve the quality of life of local communities [6].

The local housing strategy (ELH) is a tool that aims to define the intervention strategy regarding housing policy, allowing for the adaptation of the instruments defined in the new generation of housing policies (NGHP) to the territorial reality and its integrated implementation, focusing on promoting housing solutions for the most vulnerable communities. The implementation of ELH is important to respond to the guidelines of the national housing strategy (ENH) and the objectives of the "1° Direito" program, which aim to promote access to adequate housing for people living in inadequate housing situations and who do not have the financial capacity to find decent housing solutions.

Landscape resilience is a critical strategy for ensuring the safety, health, and well-being of communities, as well as maintaining the integrity of ecosystems. Landscape resilience is defined as the ability of an ecological system to adapt and recover from disturbances while maintaining its ecological functions and ecosystem services. Incorporating landscape resilience in the implementation of local housing strategies (ELH) can ensure the sustainability and resilience of local communities.

In this paper, we undertake a comprehensive analysis of housing policies and their impact on urban resilience and sustainability in Machico, Madeira. Our objective is to assess the implementation of local housing strategies (ELH) in Machico, with a specific focus on diagnosing housing needs, proposing and evaluating solutions, and projecting future housing objectives. We aim to highlight the critical interplay between housing policies, landscape resilience, and the overall well-being of the community in this context.

## 2. Materials and Methods

The "Materials and Methods" of this article encompasses the methodology used to conduct a comprehensive analysis of housing deficiencies in the Municipality of Machico, as well as the integration of these analyses with principles of landscape resilience and existing housing policies. The information presented in this section is essential to understanding how housing deficiencies were identified and assessed, how housing solutions were developed in alignment with the objectives of the "1st Right" program, and how landscape resilience was incorporated into the housing context of Machico.

First, we will describe the characterization of the housing stock in the Municipality of Machico, detailing the data sources and analysis techniques employed to gain a comprehensive overview of the housing situation. Next, we will address the analysis of housing supply and demand, including the metrics and indicators used to assess the housing needs of the community.

Subsequently, we will examine housing deficiencies and access barriers, explaining how these challenges were identified and categorized. The SWOT analysis will provide insights into existing housing policies, highlighting their strengths, weaknesses, opportunities, and threats.

Following that, we will discuss the priority lines of intervention, emphasizing how they were identified and aligned with community needs. Additionally, we will provide an overview of the current municipal housing policy, contextualizing the landscape in which our analyses and interventions are situated.

Lastly, this section will also delve into the methodology employed to promote landscape resilience in conjunction with housing solutions and offer information about the natural ecosystems in Machico, essential for our understanding of the environmental context.

These methodological details are crucial to underpinning the findings and discussions presented in the subsequent sections of this article.

The housing situation in Machico is examined, considering various typologies and demand factors. The paper explains the significance of understanding housing demand and presents an overview of the different typologies present in the region, including houses with 3 bedrooms and a living room (T3 typology). The analysis demonstrates the importance of tailored housing solutions that address the needs of diverse socioeconomic groups. In the context of Portugal, the term "T3" is specifically employed to classify a dwelling comprising three bedrooms and a living room. It is noteworthy that this terminology exclusively encompasses these specific room allocations and excludes other spatial divisions, such as living areas, kitchens, bathrooms, and storage spaces.

## 2.1. Diagnosis of Housing Deficiencies in the Municipality of Machico

The Municipality of Machico presents housing deficiencies mainly in terms of size and state of conservation of the housing units. According to on-site analysis and consultation with municipal documentation, such as the municipal master plan (PDM) and the strategic urban rehabilitation program (PERU), it is possible to observe many small and old housing units that require improvement and renovation [7]. In addition, overcrowding of housing units is also a problem, with many of them accommodating more people than recommended.

To address these deficiencies, it is important to implement local housing strategies that aim at urban rehabilitation and construction of new housing units with appropriate dimensions and suitable living conditions. It is important to mention that the implementation of these strategies should be carried out in conjunction with local communities, considering their specific needs and characteristics. These conclusions are based on on-site analysis and consultation with municipal documentation, such as the municipal master plan (PDM), the strategic urban rehabilitation program (PERU), and urban rehabilitation areas (ARU).

The evolution of the housing stock in the Municipality of Machico is closely linked to the geographical, demographic, and economic dynamics of the territory. The municipality, located in the Autonomous Region of Madeira, has an area of 68.31 km$^2$ and a population of 21,828 inhabitants [7], divided into 5 parishes. The climate and terrain vary between the coast and the interior of the municipality, with a warmer and drier climate on the coast and a cooler and more humid climate in the interior, where there is also a greater variety of vegetation. The economy of the municipality is predominantly based on the tertiary sector, including agriculture, livestock, and fishing. The population of the municipality has experienced constant growth since the end of the 19th century and mid-20th century, representing an increase of 116% since then. However, there was a 3% decrease in the population in the 1960s (Table 1). The evolution of the housing stock should be analyzed considering these demographic, economic, and geographic aspects.

According to data from the National Institute of Statistics (INE) [8], the Municipality of Machico is composed of 9954 classic family dwellings, with an average age of 31.2 years, of which 41% of classic family dwellings are 40 years or older. The proportion of very degraded buildings increased from 0.9% in 2001 to 2.51% in 2011, indicating a deteriora-

tion in the state of the conservation of the housing stock. When including buildings in need of major repairs, the proportion is 5.97%. There is a significant number of vacant buildings in the municipality, 13.17% (1297 dwellings) and seasonal occupancy dwellings, 13.15% (1295 dwellings). The Machico parish has the highest number of classic family dwellings (51%), as well as the largest number of residents, representing 52% of the total population of the municipality [8]. Cases of housing deprivation are dispersed throughout the municipality, with the highest percentage concentrated in the Machico parish. However, the municipality has strong economic and social cohesion due to its wealth of natural resources, tourism potential, traditions, and investments made by the municipality in urban requalification and sustainability.

**Table 1.** Evolution of the number of family dwellings and resident population [8].

| Year | Family Dwellings | Resident Population |
| --- | --- | --- |
| 1960 | 4780 | 21,606 |
| 1970 | 4430 | 21,010 |
| 1981 | 5288 | 22,126 |
| 1991 | 6483 | 22,016 |
| 2001 | 7363 | 21,747 |
| 2011 | 9851 | 21,828 |
| 2021 | 9954 | 19,594 |

Regarding new constructions carried out between 2011 and 2020, according to data from the INE, the T3 typology stands out the most, representing 65% of new constructions. This indicates that there was an interest in building larger houses and apartments, which may indicate a need for larger dwellings in the region (Figure 1). T0, T1, T2, T3... are the terms used to designate a dwelling with 0, 1, 2, 3... bedrooms and a living room. In other words, other divisions such as living rooms, kitchens, bathrooms, and storage spaces are not considered in this terminology.

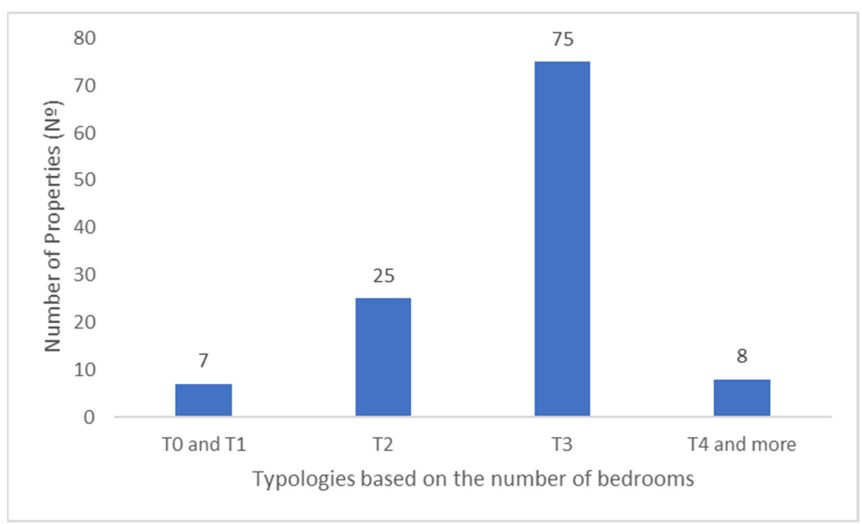

**Figure 1.** Completed dwellings (Nº) in new constructions for family housing between 2011–2020 [8].

The analysis of the housing stock's state of conservation in Machico aimed to identify and observe instances of disadvantaged housing within the municipality. The characterization of the housing stock followed the parameters mandated by the local housing strategy and subsequent programs. Through sampling and on-site observation, we identified several housing complexes with typologically and/or morphologically cohesive characteristics. These housing units, even though situated in different territorial contexts, share common features such as the prevalence of single-family buildings owned by private individuals, characterized by poor construction quality.

Regarding the classification of the state of conservation of the observed private single-family housing, it spans from "Poor" to "Very Poor". Unfortunately, the specific conditions that correspond to each classification, such as what constitutes a "Poor" condition, were not clearly defined within the provided information. These classifications are tied to construction pathologies resulting from subpar construction quality, which exacerbate issues related to unsanitary conditions, insecurity, and overcrowding.

Importantly, it is worth noting that residences falling into the "Very Poor" classification represent homes with severe housing needs. These dwellings serve as housing for vulnerable population segments, enduring unsanitary and insecure conditions. The primary goal of this analysis is to identify these situations to facilitate their addressing through the local housing strategy and subsequent programs. Further clarification of the classification scale, particularly elucidating the criteria for "Poor" conditions and others, would enhance the understanding of the housing stock's state of conservation.

The resident population appropriates space differently, with an increasingly aging population working in subsistence agriculture in the less densely populated areas. There is some dependence on private transportation for travel within the municipality, but there is a regular public transportation network of passenger buses and various routes that provide transportation between the parishes of Machico and the municipalities of Santana, Machico, and Funchal. Nevertheless, there are mobility difficulties for the most vulnerable population due to financial constraints.

A strategy to address the housing needs of the municipality of Machico must consider the specific physical, social, economic, and heritage strengths and characteristics that define the municipality's specific character to promote its territorial and social cohesion and leverage sustainable development. This includes considering the differences between the areas of the municipality, such as rural and urban areas and the specific housing needs of these areas. The strategy should encompass measures to maximize the use of existing resources, such as the rehabilitation and revitalization of existing heritage, rather than building new housing in areas of high landscape value. Additionally, the strategy should include measures to improve accessibility and mobility to ensure that people can access essential goods and services regardless of their location. It is also important to involve measures to strengthen social connections, keep families in their communities of origin, and reinforce and consolidate the morphological and landscape unity of the different nuclei that make up the municipality's identity.

Figures 2–6 present representative examples of some properties in the parishes of Água de Pena, Caniçal, Machico, Porto da Cruz, and Santo António da Serra in the context of the housing stock visit, where precarious constructions, unsanitary conditions, pathologies associated with humidity, among others, were observed.

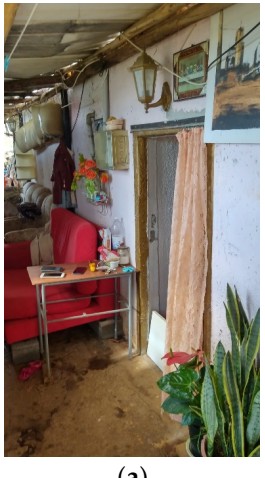 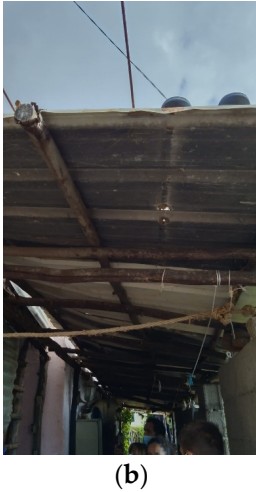

(**a**)                                              (**b**)

**Figure 2.** Building unsuitable for residential use in the parish of Água de Pena: (**a**) main entrance; (**b**) damaged roof.

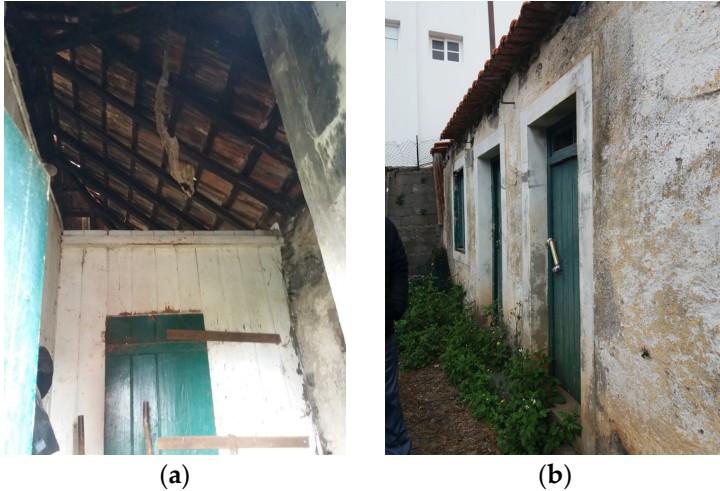

**Figure 3.** Building incompatible with residential use in the parish of Caniçal: (**a**) damaged roof; (**b**) main door.

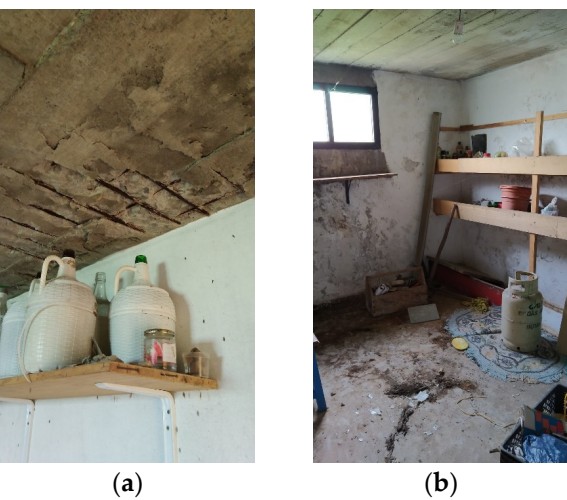

**Figure 4.** Building incompatible with housing use in the parish of Machico: (**a**) concrete slab with visible rusted reinforcement; (**b**) interior space with humidity on the walls.

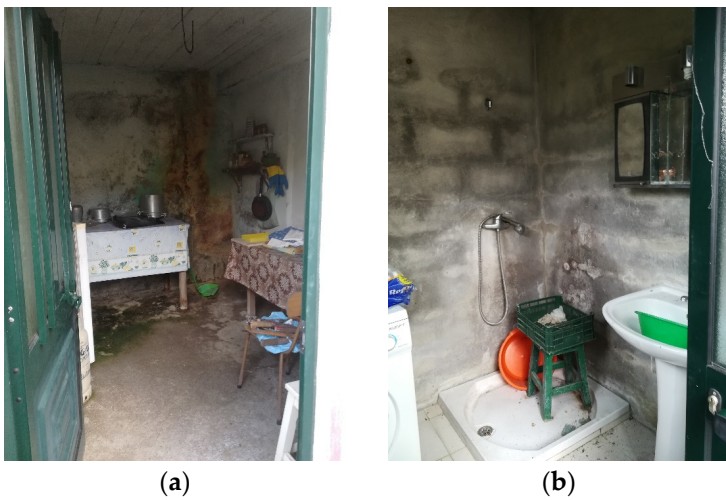

**Figure 5.** Building incompatible with residential use in the parish of Porto da Cruz: (**a**) interior space with humidity on the walls; (**b**) sanitary installation with several infiltration and humidity problems.

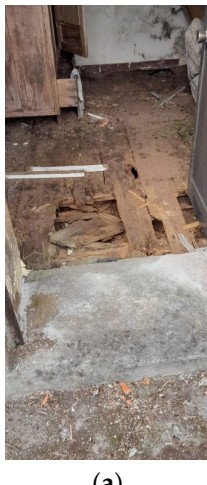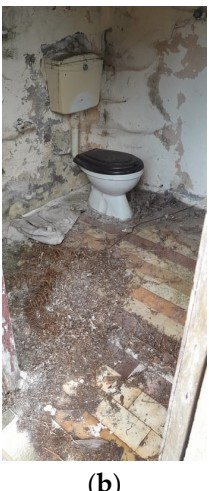

(**a**)          (**b**)

**Figure 6.** Building not suitable for residential use in the parish of Santo António da Serra: (**a**) ground floor with rotten wooden flooring; (**b**) sanitary installation with several infiltration and humidity problems.

In our quest for progress, it is of paramount importance to evade the blunders of the past and veer away from strategies that yield dissatisfaction, both among builders and residents. The recurrent inclination toward minimal residential spaces, occasionally falling short of the benchmarks stipulated by the general regulation of urban buildings (RGEU), sanctioned by Decree-Law No. 38 382 of 7 August 1951, has been responsible for predicaments such as overcrowding, hastened wear and tear of housing structures, and the erosion of quality of life and well-being. The unyielding rigidity in internal layouts, spurred by concerns of spatial optimization and financial prudence, hinders the adaptability to the ever-evolving requirements of families and demographic shifts, demanding costly interventions for adjustments.

Furthermore, the dearth of quality and thermal comfort in constructions, a byproduct of a short-term economic outlook fixated solely on construction expenses rather than the long-term cost-to-benefit ratio, has proven to be wanting. This shortfall not only impacts the health and well-being of residents but also presents challenges for their social reintegration and re-entry into the job market. The gravity of the situation is further magnified in Machico due to the inadequacy of construction quality and the pressing need to meet energy efficiency standards.

Harmonious integration within the locale and symbiotic linkage with public spaces also pose concerns. The rugged terrain and concentration in specific areas can give rise to mobility challenges and isolation predicaments, particularly for vulnerable and elderly populations. As we navigate the intricacies of housing strategies, it is imperative to factor in the notion of basic housing, ensuring that the foundations of habitation encompass the essentials for dignified living.

The municipality of Machico, located in the Autonomous Region of Madeira, has a population density above the national average, with 19,594 inhabitants distributed over a territory of 68 km$^2$. Most of the population is concentrated in the parish of Machico and its valley, where the city and the county seat are located. Another important population center is Caniçal due to the presence of the commercial port, which is the main gateway for goods to the island of Madeira and the Madeira Free Zone. The municipality also has a connection to Madeira airport, generating significant economic activity. In terms of landscape and environment, the municipality has a significant part of the Madeira Natural Park, including the Laurissilva Forest, a UNESCO heritage site since 1999, and the Santo da Serra golf club. In economic terms, although the tertiary sector is predominant, the construction sector also has a significant weight in employment.

Analysis spanning from the first quarter (Q1) of 2016 to the fourth quarter (Q4) of 2020 reveals that housing sales prices within the municipality of Machico exhibited consistent

fluctuations, encompassing both ascents and descents. According to data sourced from Idealista, a real estate advertisements platform, as of June 2021, the highest average price per square meter for properties for sale is registered in the parish of Santo António da Serra (1978 €/m$^2$), followed by Machico (1430 €/m$^2$) and Caniçal (1366 €/m$^2$) (Idealista, 2021). Notably, the parish of Porto da Cruz exhibits the lowest average price. It is important to note that while alignment is observed between the average values per square meter from Idealista and local real estate agencies, the scarcity of properties in Porto da Cruz and Santo António da Serra has significantly impacted the average price. Figure 7 offers a comparison of the mean sales price per m$^2$ across the five parishes.

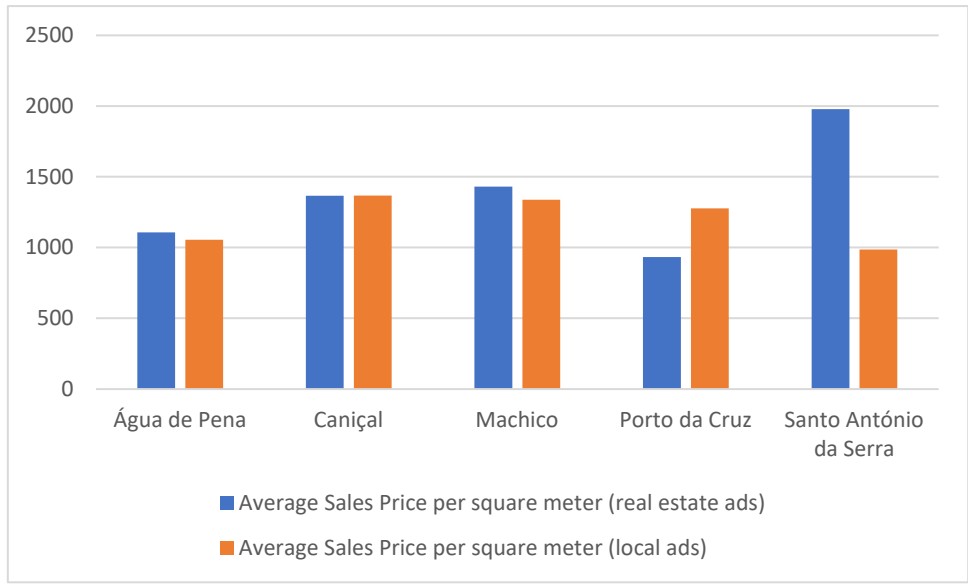

**Figure 7.** Average sales price per square meter by parish [9,10].

Analyzing data from June 2021 (Table 2), it is evident that among the 183 properties available for sale, a substantial portion is in the parish of Machico (118), while the remainder is distributed throughout other parishes within the municipality [9].

**Table 2.** Number of properties available for sale by parish [9].

| Parish | Number of Properties Available for Sale |
| --- | --- |
| Água de Pena | 31 |
| Caniçal | 13 |
| Machico | 118 |
| Porto da Cruz | 13 |
| Santo António da Serra | 8 |

Insights from Mendes underscore that "the rental market in Portugal has been grappling with an increasingly scarce supply, leading to a significant surge in rental prices" [11]. The study accentuates that the dearth of rental properties stems from elevated purchase prices, rendering market entry arduous for new buyers, coupled with a decrease in property availability due to unfavorable fiscal policies towards rentals. The dearth of rental properties, both in Machico and Portugal at large, has evolved into a mounting challenge, curtailing housing options for residents and potentially fueling demand for housing in alternate locales.

Based on data from the National Institute of Statistics (INE), the construction of new family housing units in the municipality of Machico displayed a declining trajectory between 2011 and 2020, averaging around 13 new structures per annum. This trend is visually confirmed in Figure 8, illustrating 2011 as the year with the highest number

of new constructions in the municipality (28), with subsequent years, specifically 2015 through 2018, witnessing markedly fewer constructions—4 in 2016 and 5 in 2015, 2017, and 2018. Moreover, expansions, alterations, and reconstructions demonstrated an uneven trajectory, recording figures below the period's mean between 2014 and 2017. However, 2020 showcased a convergence toward the values observed in 2011.

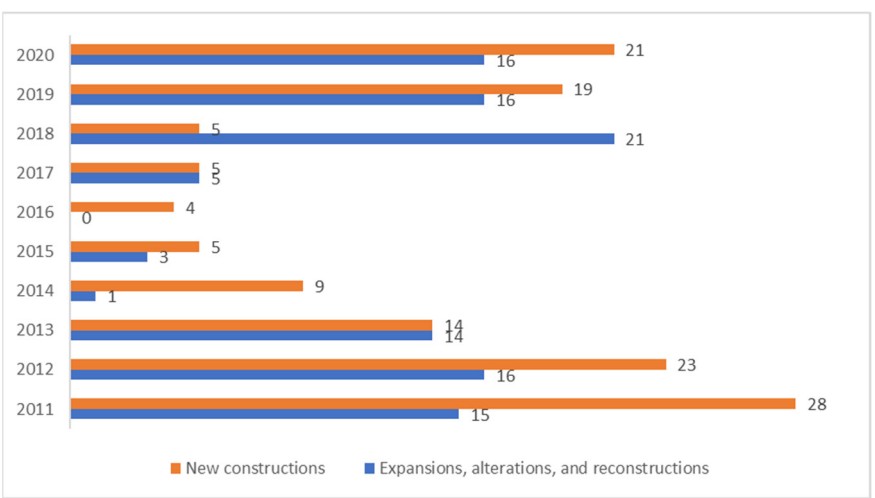

**Figure 8.** Buildings completed for family housing 2011–2020 [12].

As highlighted by the "Câmara Municipal de Machico" [13], urban revitalization stands as a paramount priority for the municipality, geared towards enhancing citizens' living conditions and bolstering regional economic growth. In pursuit of this, initiatives have been initiated to restore historic edifices, rejuvenate public spaces, and establish novel infrastructure. Furthermore, projects promoting tourism are cultivated, encompassing the revival of cultural heritage and the establishment of novel tourist facilities. The "Câmara Municipal de Machico" [13] places additional emphasis on education and training as pivotal for promoting human advancement and equitable opportunities, channeled through investments in educational infrastructure and citizen-oriented training programs.

To elevate the region's profile and attract visitors, the "Câmara Municipal" introduced the tourism portal www.visitmachico.com (accessed on 27 August 2023), serving as a key conduit for spotlighting the Machico region and the Autonomous Region of Madeira. This digital venture not only facilitates regional promotion but also facilitates the gathering of "digital experience" and evaluation of visitor behavior through annual analysis of search metrics conducted by the municipality.

In June 2021, the Idealista real estate ads platform listed 183 properties available for purchase within the municipality of Machico, reporting an average sales price of €1363 per square meter. Pertaining to the rental market, a mere 9 properties were identified as available for rent, all situated in the Machico parish, carrying an average value of €8.65/m$^2$.

When analyzing the available offer in the municipality of Machico in June 2021, it becomes apparent that a significant proportion of properties for sale are categorized as T3 typology (43%). Properties boasting T4 or greater typologies constitute 23% of the inventory, T2 typologies constitute 20%, and T1 typologies constitute 14% of the available stock. Evidently, the available offerings appear well-equipped to cater to the requisites of the prevailing socioeconomic groups in Machico, as substantiated by the average family size of 3.0 individuals within the municipality [8]. The dominant family nuclei in the region are primarily composed of couples with one child (41%), followed by couples with two children (28%), childless couples (24%), and families with three or more children (8%) (Figure 9).

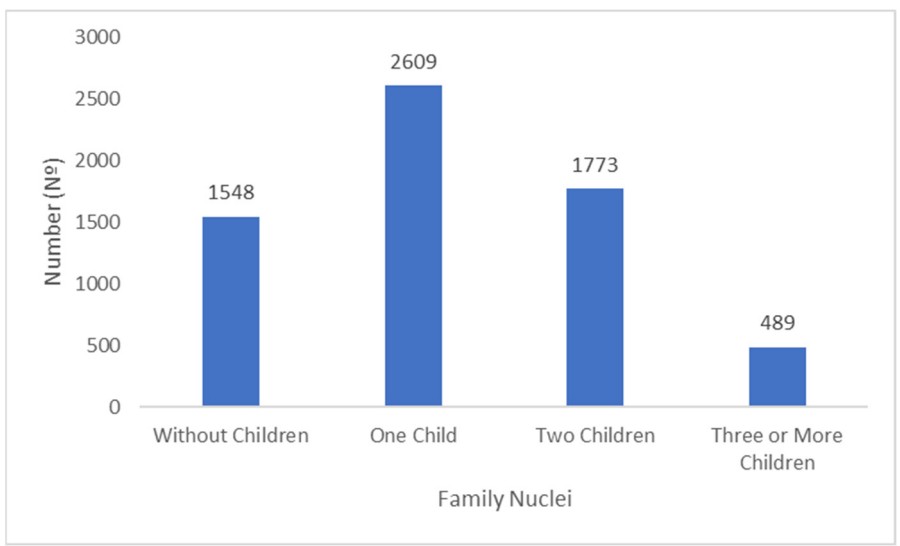

**Figure 9.** Family nuclei by typology 2011 [8].

Table 3 furnishes insights into the average price per m$^2$ for houses for sale, segmented by typology.

**Table 3.** Price per m$^2$ of houses for sale by typology [8].

| Typology | Average Price €/m$^2$ | Number Available on Housing |
|:---:|:---:|:---:|
| T0 | | 0 |
| T1 | 1960 € | 26 |
| T2 | 1267 € | 36 |
| T3 | 1121 € | 78 |
| T4+ | 1508 € | 42 |

Upon analyzing the housing sales landscape, the T1 typology commands the highest sales prices per square meter (€1960/m$^2$), notwithstanding its relatively low availability, barring the T0 typology, which lacks any available units. Conversely, the T3 typology exhibits the greatest supply for sale within the municipality, accompanied by an average price per square meter of €1121/m$^2$, presenting the more modest values. The T2 and T4 typologies showcase values of €1267/m$^2$ and €1508/m$^2$, respectively.

The presence of youth within the confines of Machico plays an influential role in demographic dynamics, thereby impacting housing demand. A retrospective study covering the period from 2009 to 2016 reveals a decline in the number of residents aged between 20 and 34 years, reflecting a decrease of 117 individuals (Figure 10). Subsequently, from 2017 onward, a gradual and consistent upsurge ensued, culminating in an absolute increase of 30 young residents within the municipality by 2019. This demographic evolution compels the municipality to adopt measures to fulfill the housing needs of this specific cohort, intent on achieving independence, establishing families, or embracing solitary living.

It is apparent from the presented figures that the sustained drop in Machico's population corresponds with outward migration, as highlighted by migratory balance. However, the housing demand witnessed in recent years could be attributed to the surge in youth yearning for autonomy, coupled with the rise in young residents since 2017, even though the augmentation remains incremental. Acknowledging the scarcity of housing supply in the municipality, particularly within the rental market, prospects for youth and families remain restricted. In parallel with recent territorial investments, particularly in the social, educational, and tourism sectors, supplemented by the support provided during the COVID-19 pandemic, real estate sector investments assume critical importance in retaining the population, particularly the young populace in pursuit of novel employment avenues.

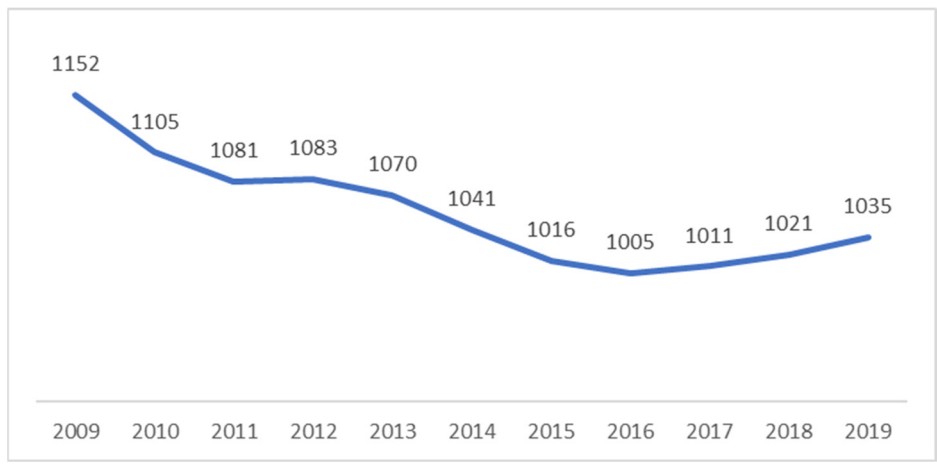

**Figure 10.** Evolution of the number of young people aged 20–34 in Machico between 2009–2019 [12].

To compute the monthly expenses linked to purchasing a house in Machico through 90% financing, a simulation of real estate credit was undertaken for a 30-year tenure in July 2021. Based on the available inventory, the median monthly cost for a T1 apartment priced at €110,000 tallies at €319.15, for a T2 valued at €137,000 it stands at €397.49, for a T3 valued at €185,000 it equals €536.75, and for T4 or larger configurations priced at €389,000, it reaches €1128.32. Considering the median gross income reported by taxpayers in Machico and deduced from income tax amounts to €8479 annually (INE, 2021), which was then divided by 12, the median monthly income of those employed in the municipality is €706.58.

According to standard banking practices, the debt-to-income ratio establishes that housing expenses should not exceed 30% of a household's monthly income. When considering the prevailing costs of real estate, a household with a single adult, earning the median gross income derived from tax calculations, would not meet the criteria for obtaining bank financing for any type of housing (as indicated in Table 4). Similarly, households consisting of two adults, earning the median monthly income determined from tax data, would not qualify for a 30-year bank loan with a debt-to-income ratio of 30% or lower for T3 and T4 housing types (as shown in Table 4).

Examining the family composition in the municipality based on the 2011 census, most families are composed of couples with one or two children. Single-parent families make up 16% of the total family structures in the municipality, with 86% of these single-parent households being led by mothers with children [8].

**Table 4.** Access to housing for purchase by household type [13].

| Type of Household | Can Access the Buying/Selling Market | Affordability Ratio–Buying/Selling |
|---|---|---|
| Unipersonal | T1—NO | 45% |
| Single parent | T1—NO | 45% |
| | T2—NO | 56% |
| | T3—NO | 76% |
| Couple without children | T1—YES | 23% |
| | T2—NO | 28% |
| | T3—NO | 38% |

**Table 4.** *Cont.*

| Type of Household | Can Access the Buying/Selling Market | Affordability Ratio–Buying/Selling |
|---|---|---|
| Couple with 1 child | T2—YES | 28% |
| | T3—NO | 38% |
| | T4—NO | 80% |
| Couple with 2 children | T2—YES | 28% |
| | T3—YES | 38% |
| | T4—NO | 80% |
| Couple with 3 or more children | T3—NO | 38% |
| | T4+—NO | 80% |

It has been identified that 52 families live in insufficient housing conditions, with most of them concentrated in the parish of Machico, representing 58% of the total number of families with housing needs. These households face a range of problems, such as poor sanitation and insecurity, which are a priority to be addressed by the municipality. Table 5 shows the distribution of housing needs by parish.

**Table 5.** Distribution of housing deprivation situations by parish [13].

| Parishes | Families | People |
|---|---|---|
| Água de Pena | 5 | 10 |
| Caniçal | 10 | 31 |
| Machico | 30 | 91 |
| Porto da Cruz | 6 | 11 |
| Santo António da Serra | 1 | 2 |

The Machico parish is the main one in the municipality of Machico, with 52% of the resident population. Despite being a central area with various facilities and good accessibility, there is a higher percentage of families in housing need. The Caniçal parish is the second with the highest number of residents and needy families, accounting for 19% of the identified cases. On the other hand, the Santo António da Serra parish has the least cases of housing need, with only 2% of the identified cases. This parish has a lower population density and faces challenges of accessibility and mobility. Regarding the type of occupancy, the families identified as having housing need are divided between those who own their homes (75%) and those who rent (25%).

Housing need in the municipality of Machico results from various factors, such as difficulty in accessing the housing market due to the inflation of prices; scarcity of supply, mainly in rental housing; deterioration of the buildings; the financial incapacity of residents to rehabilitate dilapidated housing, especially in peripheral parishes; households with low financial resources; unsanitary and insecure housing; the aging population; unsanitary or inadequate housing in extended households; lack of rental housing supply and attractiveness in rural areas; financial and social problems due to lack of employment and the COVID-19 pandemic; family management and mental health problems.

The municipality of Machico has 52 families identified as being in a situation of housing deprivation. According to Table 6, many members of these households are retirees or pensioners (43 members, or 30%), followed by students (25%). The condition of being employed is the third most common, with 22% of members. It is important to highlight that these workers usually have low professional qualifications and monthly income, which reflects the financial situation of these families. In all, 16% of members are unemployed, and there is also 1 person characterized as a domestic worker and 1 self-employed worker. A total of 6% of members do not report their employment status. The local development

strategy for 2014–2020 shows that the number of recipients of unemployment benefits increased by 40% between 2007 and 2013. There was an 11% increase in the number of pensioners in that period, representing 33% of the active population. In addition, the aging of the population is a growing trend in the municipality. The most common level of education is the 1st cycle of basic education, according to the 2011 census. Other levels of education also show increases compared to 2001, except for the 2nd cycle of basic education. These data indicate that housing deprivation in the municipality of Machico is related to a combination of factors, including the income and financial condition of families, as well as the scarcity of affordable housing supply and the employment status of individuals.

**Table 6.** Situation regarding employment of the members of the signaled families [13].

| Employment | Absolute Value | Relative Value |
|---|---|---|
| Student | 36 | 25% |
| Retired/Pensioner | 43 | 30% |
| Employee (salaried worker) | 32 | 22% |
| Unemployed | 23 | 16% |
| Domestic worker | 1 | 0.50% |

Table 7 presents a breakdown of different family typologies that are experiencing housing deprivation within the municipality of Machico. These family typologies encompass various compositions and structures. The term "nuclear households with children" refers to families consisting of parents and their children (34% of cases). "Extended households" denotes families that include additional relatives beyond the nuclear family, such as grandparents, aunts, uncles, or cousins (21% of cases). "Isolated households" indicate families where individuals live alone or with unrelated individuals (23% of cases). "Single-parent households" pertains to families headed by a single parent (12% of cases). Lastly, "nuclear households without children" denotes families consisting of parents without dependent children (10% of cases). This analysis underscores that various family structures are encountering housing challenges in Machico.

**Table 7.** Signaled Family Typology [13].

| Aggregated Typology | Number of Families | Number of People |
|---|---|---|
| Families with children | 18 | 66 |
| Extended families | 11 | 45 |
| Isolated families | 12 | 12 |
| Single-parent families | 6 | 12 |
| Nuclear families | 5 | 10 |

Referring to Table 8, the comprehensive analysis of effort rates (T.E.) among the families identified within the Machico municipality underscores a prevailing trend where most of these families exhibit effort rates exceeding 30%. This phenomenon suggests a distinct financial imperative for these families when it comes to engaging with the rental market. Notably, an effort rate surpassing 30% indicates that the average gross monthly income of these families falls below four times the indexante dos apoios sociais (IAS), which stands at 1755.24 €. This regulatory yardstick, introduced by the Portuguese state on 29 December 2006 through Law No. 53-B/2006, seeks to establish a framework for diverse social support mechanisms.

Furthermore, this comparative evaluation considers rental market prices within the Machico municipality during the latter half of 2020. The analysis focuses specifically on the median cost per square meter ($m^2$) of T3 typologies, which is indicative of the prevailing pricing trend for the most abundant housing supply in the area. To clarify, the "Max. rent" column in Table 8 is calculated based on the given maximum effort rate of 35% for

affordable rent. It represents the maximum monthly rent that a family with a specific income level can afford, given the specified effort rate.

**Table 8.** Comparative Analysis of Effort Rates (T.E.) [13].

| Income | Monthly Rent 186 m$^2$ 3.95 €/m$^2$ T3 | Rent Support | | Social Housing | | Affordable Rent | |
|---|---|---|---|---|---|---|---|
| | | T. E | Max. Rent | T. E | Max. Rent | T. E | Max. Rent |
| 1 IAS | 438.81 € | 734.70 € | 23% | 100.22 € | 25% | 108.94 € | 35% | 152.52 € |
| 2 IAS | 877.62 € | 734.70 € | 23% | 200.45 € | 25% | 217.88 | 35% | 305.03 € |
| 2.5 IAS | 1097.03 € | 734.70 € | 23% | 250.56 € | 25% | 275.35 | 35% | 381.29 € |
| 3 IAS | 1316.43 € | 734.70 € | 23% | 300.67 € | 25% | 326.82 | 35% | 457.55 € |
| 3.5 IAS | 1535.84 € | 734.70 € | 23% | 350.79 € | 25% | 381.29 | 35% | 533.81 € |
| 4 IAS | 1755.24 € | 734.70 € | 23% | 400.90 € | 25% | 435.76 | 35% | 610.06 € |

With these refinements, it becomes evident that the families identified within the municipality of Machico are indeed facing a palpable financial need when attempting to access the rental market.

Table 9 shows that the effort rate is significantly high for households with lower incomes, given the available rental supply in the municipality of Machico. Specifically, for those whose average monthly income is equal to or less than 433.25 €, the effort rate is almost 80%, which means that these families would be spending almost all their available income on rent. This data reflects the difficulties that low-income families face in accessing the rental market, especially considering the scarcity of available supply in the municipality.

**Table 9.** Effort rates by income level and the most available typology [13].

| Income | Median Rent T3 |
|---|---|
| 1 IAS | 167% |
| 2 IAS | 83% |
| 2.5 IAS | 67% |
| 3 IAS | 56% |
| 3.5 IAS | 48% |
| 4 IAS | 42% |

The comparative analysis of the effort rates shows that the housing shortage in the municipality of Machico is aggravated by the scarcity of available rental properties. The lack of affordable housing mainly affects households with lower incomes, which fall below the limit of 4 times the IAS. The inability to access the rental market with adequate effort rates reflects the difficulty these families face in meeting their housing needs.

As illustrated in Figure 11, it is possible to observe that the variation in the price per square meter for sale and rent is growing at a faster pace than the variation in the monthly income of employees. This indicates that the financial capacity of families is becoming increasingly lower in relation to access to the real estate market, especially about rental properties. Additionally, the trend of mismatch between price and income variations makes the situation even more difficult for families in need, as they are unable to access bank credit.

The analysis of the distribution of families in housing need in the municipality of Machico reveals that the parish of Machico has the highest concentration of cases, with 58% of families so identified. This parish's central location and privileged accessibility contribute

to easier access to major facilities and services, including the job market. The parish of Caniçal comes in second place, with 19% of cases, due to its low-income characteristics and larger household size. The parishes of Água de Pena and Porto da Cruz account for 12% and 1% of the cases, respectively. The housing needs of the identified families in the more peripheral parishes of the municipality are also influenced by their location, accumulating housing vulnerabilities and a higher risk of exclusion due to the distance from facilities and services, as well as the scarcity of public transportation.

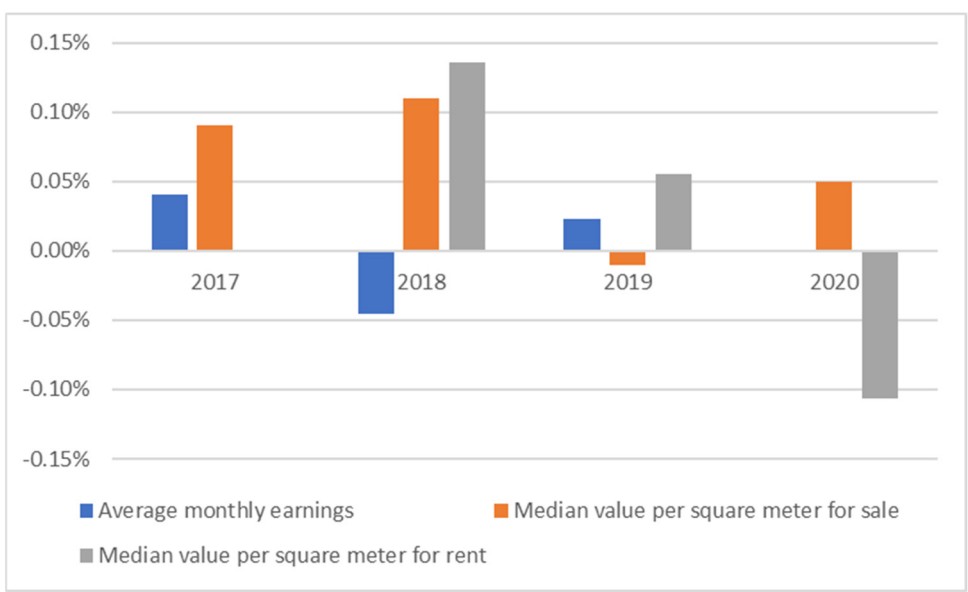

**Figure 11.** Comparison of the variation in the price per square meter for sale and rent with the variation in the monthly earnings of salaried workers [12].

The situations of indecent housing identified in the study were classified according to the degree of urgency. The first situation is that of unhealthiness and insecurity, which is mainly due to the advanced degradation of the buildings, facing problems with roofs, walls, thermal insulation, infiltration, and humidity, causing impacts on the health of residents. In addition, these housing units also have an absence of or insufficient coverage, bathroom, sanitation, and heating/thermal insulation. The second situation is that of inadequacy, which stands for the existence of architectural barriers in housing, especially for households with members with disabilities and the elderly population.

According to Table 10, families requesting housing support are those facing financial problems, with most requests coming from homeowners who are unable to rehabilitate their homes. The analysis of households identified as in need reveals that the majority are living in unhealthy and unsafe conditions, representing 95% of cases eligible for the "1º Direito" program. In addition, 13 flagged families face difficulties in the private rental market and do not meet the requirements to be eligible for the "1º Direito" program, requiring support in rental access programs. It is important to consider the sociodemographic evolution of the municipality and the needs of its inhabitants to develop housing solutions that meet their needs and contribute to the social and economic development of the territory.

**Table 10.** Main housing needs situations according to reported cases [13].

| Situation of Housing Need | Occupancy Regime | Number of Families | Number of People |
| --- | --- | --- | --- |
| Unhealthiness and Insecurity | Owner | 37 | 66 |
| Inadequacy | | 2 | 45 |
| Not eligible | | 12 | 13 |

To comprehensively evaluate the real estate sector in the municipality of Machico, a SWOT analysis was conducted, involving a diverse range of stakeholders. This analytical framework delved into the strengths, weaknesses, opportunities, and threats inherent in the local real estate market. Participants in this analysis included citizens, public administrators, local real estate agencies, and relevant community organizations.

The SWOT analysis serves as a powerful tool to synthesize the inherent advantages and challenges faced by the real estate sector in Machico. By systematically assessing the landscape, this approach facilitated the identification of strategies and objectives to address limitations and capitalize on existing prospects. The insights derived from this analysis are pivotal in formulating a robust local housing strategy, aimed at enhancing the social and urban well-being of the territory.

Considering the presented challenges and potential threats, a prudent approach entails harnessing the recognized strengths. These strengths serve as cornerstones for the development of the local housing strategy, with the goal of meeting the pressing needs of the community. Notably, the comparative affordability of property prices in Machico compared to neighboring municipalities, coupled with recent economic growth, an upsurge in new businesses, a rising demographic of individuals aged 20 to 34, and significant investments in the tourism sector, underscore the vibrancy of the municipality's real estate market.

Furthermore, investments in social and accessibility infrastructures emerge as drivers for improving residents' quality of life. The convergence of these factors, along with available financial incentives, holds the potential to invigorate territorial development and ameliorate the housing scarcity that families in need face.

However, contextualizing the strengths within the larger landscape is crucial. The aging housing inventory, the presence of unoccupied properties, and diminished construction activities contribute to a constrained rental market and a diminished supply of available homes for purchase. Escalating property prices have exacerbated housing accessibility issues for numerous families. Moreover, the uneven distribution of the population, primarily concentrated in the parish of Machico and scattered throughout the municipality, complicates the establishment of coherent urban centers. This dispersal also underscores the reliance on private transportation for intra-municipal travel, a challenge predominantly faced by the more vulnerable segments of the population.

It is imperative to underscore that the municipality currently lacks a dedicated social housing infrastructure to cater to the identified needs. Given these dynamics, the formulation of a robust and effective housing strategy emerges as a fundamental imperative for advancing the social and urban evolution of the territory.

By engaging with this multifaceted SWOT analysis, involving both community representatives and professionals, the study aims to facilitate informed decision-making and strategy formulation that can lead to meaningful changes and improvements in the local real estate landscape.

Based on the diagnosis carried out, some of the intervention priorities are:

- Rehabilitation of properties by owners, aiming to improve housing conditions;
- Rehabilitation of degraded or deteriorating urban areas, with the objective of protecting and enhancing the historical and cultural heritage;
- Mobilization of the owners of vacant or abandoned properties, to negotiate their rehabilitation and make them available for affordable rent, boosting the rental market in the municipality;
- Promotion of controlled-cost housing for the young population, either for sale or rent;
- Qualification of particularly vulnerable urban areas and improvement of internal mobility, aiming for inclusion and social cohesion.

Among the main objectives established in the municipal master plan (PDM) and the urban rehabilitation strategic plan (PERU) of the Municipality of Machico, the following stand out, which have direct implications for housing:

- Ensure coherent, sustainable, and innovative urban concepts;
- Assess the necessary conditions for revitalization and rehabilitation operations;

- Revitalize the historical spaces of the Municipality of Machico, preserving their memory and identity and enhancing their revitalization;
- Reverse the trend of degradation of old or historical urban spaces (EUAH), promoting the regeneration and rehabilitation of the built heritage and public space;
- Integrate the EUAH into the city and town and articulate them with their neighborhoods;
- Economically and socially revitalize the delimited areas;
- Promote environmental sustainability and energy efficiency;
- Consolidate Machico as a tourist destination through the strengthening and improvement of spaces dedicated to sports.

The promotion of social justice and territorial cohesion necessarily requires addressing identified housing needs, closely linked to situations of unhealthiness and difficulties in accessing the housing market. Therefore, housing rehabilitation solutions for homeowners are prioritized, as is raising awareness among owners of vacant properties to rehabilitate them and provide housing supply in the municipality, especially in the affordable rental market. It is important that owners of vacant properties in the municipality can benefit from specific special conditions aimed at tenant safety and benefit, especially in buildings in the urban rehabilitation areas of the municipality. Despite the various forms of support available to citizens in the field of housing, given the number of families in housing scarcity situations and the lack of social housing supply managed by the municipality, it may be inevitable to resort to the promotion of new housing, and it is essential to maintain rent supports, having identified financial difficulties for several households in terms of payment. In the housing solutions to be promoted, it will be essential to follow the guidelines of the municipal master plan to meet the needs of the municipality, promoting territorial equity while considering the needs and desires of families that need to be relocated. The quality of construction, the sustainability of materials, and the promotion of new investments in the territory are also fundamental aspects to be considered. In addition, it will be important to promote the rehabilitation of housing for owners, as well as raising awareness among owners of vacant properties for rehabilitation and housing supply in the municipality, including the affordable rental market. Despite the various housing incentives available, it may be necessary to build new housing to meet the needs of housing scarcity, while maintaining rent incentives to help families facing financial difficulties.

The PDM (plano diretor municipal) and PERU (programa estratégico de reabilitação urbana) programs play integral roles in shaping urban development strategies within the municipality of Machico. These programs underscore the importance of landscape resilience by incorporating specific provisions and measures to enhance the adaptability and sustainability of the urban environment in the face of various challenges.

The PDM, as the municipal master plan, encompasses a comprehensive vision for the spatial organization and development of the territory. Within this framework, the concept of landscape resilience is explicitly addressed by promoting sustainable land-use practices, green infrastructure development, and the preservation of natural assets. The integration of ecological protection zones and green spaces is indicative of the PDM's commitment to enhancing the municipality's capacity to withstand environmental pressures, mitigate risks, and promote long-term ecological balance.

Likewise, the PERU program, which focuses on urban rehabilitation and revitalization, acknowledges the significance of landscape resilience in achieving holistic urban transformation. This program incorporates strategies that prioritize the restoration and integration of natural elements, thereby bolstering the overall resilience of urban areas. By advocating for sustainable construction practices, the adoption of renewable materials, and the enhancement of green spaces, the PERU program aligns itself with the principles of landscape resilience. These measures not only contribute to the environmental quality of urban spaces but also augment the capacity of the community to cope with unforeseen challenges.

In essence, both the PDM and PERU programs acknowledge the inseparable link between landscape resilience and the sustainable development of the municipality. By fostering a harmonious coexistence between built and natural environments, these programs

exemplify the municipality's commitment to nurturing urban spaces that are not only adaptable and resilient but also conducive to the well-being and prosperity of its residents.

### 2.2. Landscape Resilience

The paper establishes a cohesive link between housing policies and landscape resilience. This linkage is founded on the recognition of landscape resilience's significance in urban areas, particularly in the context of housing policies. By accentuating the necessity of integrating environmental considerations into housing strategies, the paper highlights how landscape restoration plays a pivotal role in strengthening urban resilience. The discussion delves into the proactive impact of landscape restoration and the creation of green spaces on enhancing overall urban resilience.

The quality of life of the population and the environmental sustainability of the territory. In the case of Machico municipality, in Madeira, landscape resilience must be considered considering the region's vulnerability to extreme weather events such as floods [14]. To promote landscape resilience, it is essential to encourage the participation of the local community in initiatives to recover and protect the environment, as well as to increase awareness of the importance of preserving natural resources for the long-term sustainability of the community [15].

To promote landscape resilience, it is necessary that local housing strategies consider landscape resilience, ensuring the construction and rehabilitation of housing in appropriate areas and considering the characteristics of the territory, such as relief, vegetation, and hydrography. In addition, it is important to implement sustainable land management practices and green infrastructure (Figure 12).

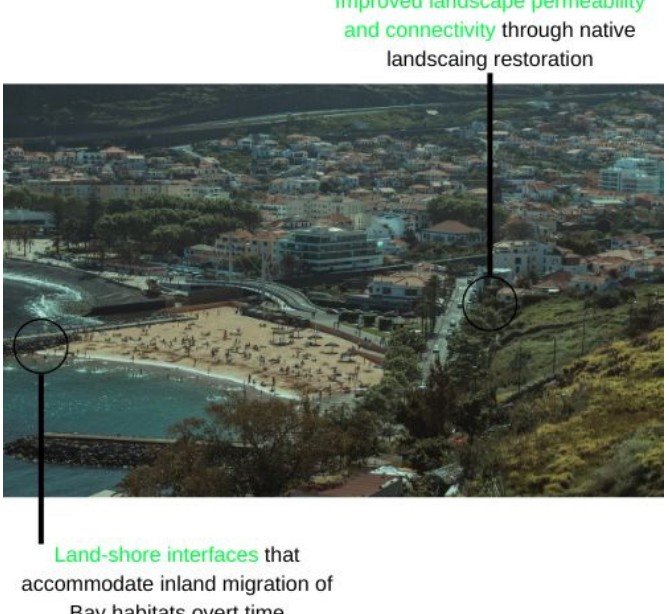

**Figure 12.** Examples of resilience objectives of Machico Bay.

The implementation of various measures such as the creation of ecological protection zones, the adoption of sustainable construction techniques, the promotion of environmental conservation practices, the establishment of green infrastructure, and the mitigation of disaster risks is crucial for enhancing the resilience of the landscape [16]. These measures, while important for landscape preservation, can also have a direct impact on housing recovery efforts. For instance, incorporating sustainable construction practices can not only contribute to landscape resilience but also improve the quality and durability of housing structures. Similarly, the establishment of green infrastructure can offer multifaceted benefits, including enhanced housing aesthetics and improved stormwater management.

Moreover, the reduction in disaster risks, as mentioned, is interconnected with ensuring the safety and longevity of both the natural environment and housing. It is imperative to recognize that these landscape-oriented strategies can complement and reinforce actions directed towards housing recovery, fostering a holistic approach to urban development [17].

Another relevant measure is the creation of environmental protection areas, such as natural parks and ecological reserves [18]. These areas can help preserve natural ecosystems and ensure the maintenance of ecosystem services that are important for the population. It is important that these areas are managed sustainably and that environmental conservation measures are implemented to prevent degradation and loss of biodiversity (Figure 13).

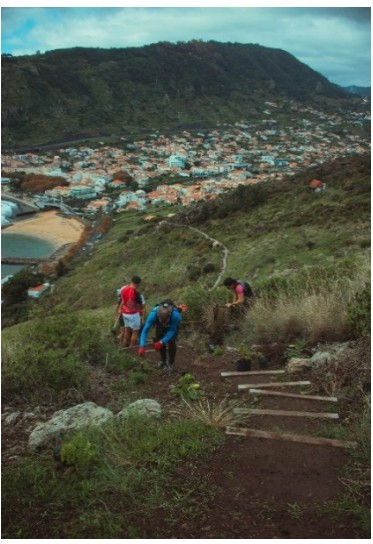

**Figure 13.** Plantation of Echium Candicans and Argyranthemum Pinnatifidum in the parish of Machico.

Promoting landscape resilience in Machico can also be achieved through the adoption of sustainable construction practices. It is important that buildings are designed to minimize the risks of natural disasters and to reduce environmental impact, for example, using materials and construction techniques that are less harmful to the environment and that reduce energy and water consumption (Figure 14).

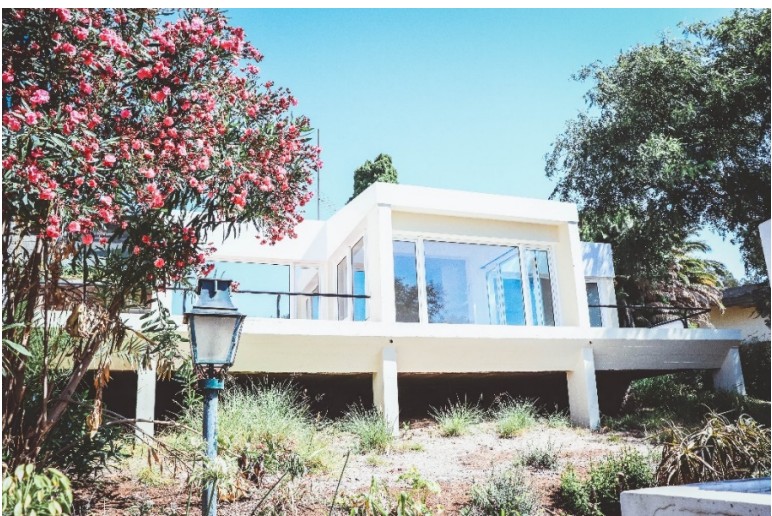

**Figure 14.** Sustainable house integrated into nature in the parish of Água de Pena.

Landscape resilience is essential to ensuring an adequate quality of life for local communities and the long-term protection of the landscape. Therefore, it is essential to implement policies aimed at the conservation and recovery of natural ecosystems,

sustainable management of natural resources, and the mitigation of disaster risks. In the municipality of Machico, Madeira, the vulnerability of the region to extreme weather events makes it even more important to adopt measures that aim to conserve and recover natural ecosystems, sustainable management of natural resources, and reduce disaster risks. Local housing strategies should consider landscape resilience, ensuring the construction and rehabilitation of housing in suitable areas, considering the terrain, vegetation, and hydrography characteristics of the area (Figure 15).

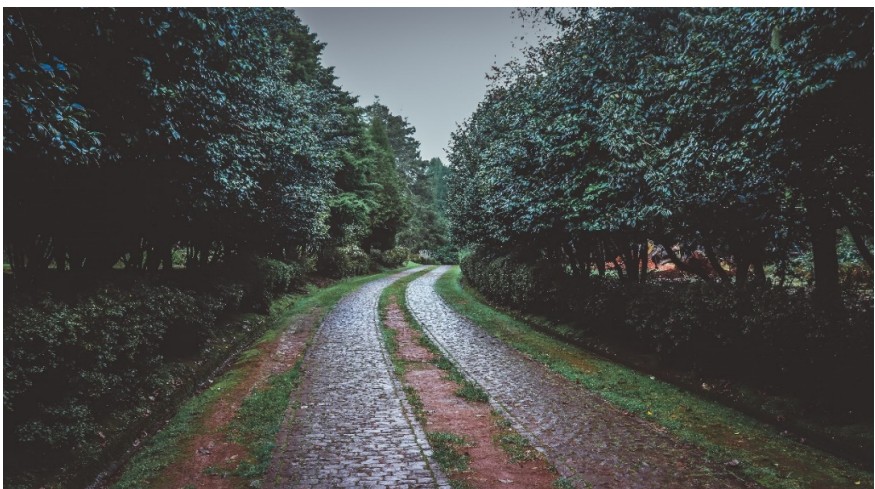

**Figure 15.** Pedestrian path perfectly integrated into nature in the parish of Santo António da Serra.

To ensure the evolution of the housing stock promoting landscape resilience, integrated and sustainable urban planning is necessary, which considers not only housing needs, but also the environmental and social aspects of the territory.

Landscape resilience is a critical element in ensuring the quality of life of the population and the environmental sustainability of the territory (Figure 16). In the Municipality of Machico, it is essential to consider landscape resilience, given the region's vulnerability to extreme weather events such as maritime swell and storms [19].

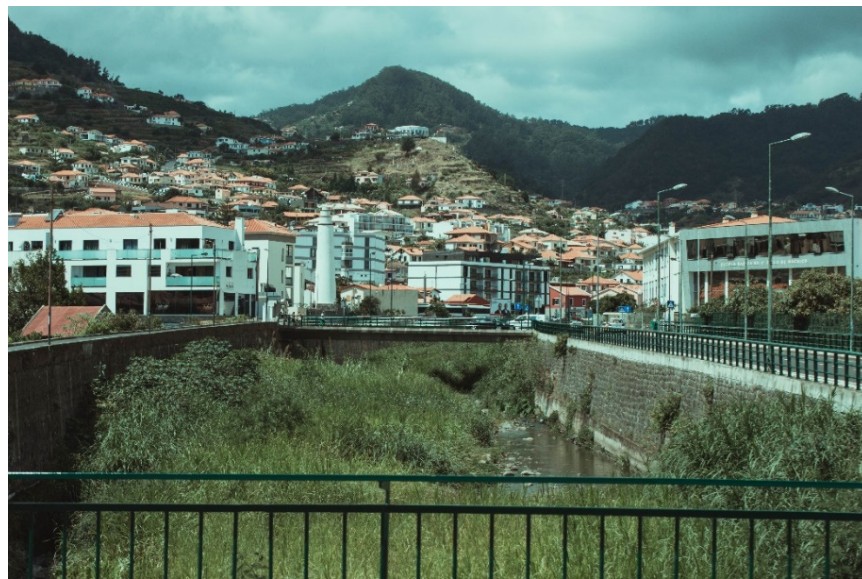

**Figure 16.** Machico river covered with natural vegetation, a habitat for several species of birds.

According to Leal Filho [20], it is necessary to adopt policies aimed at the conservation and recovery of natural ecosystems, sustainable management of natural resources, and the reduction in disaster risks to increase landscape resilience (Figure 17).

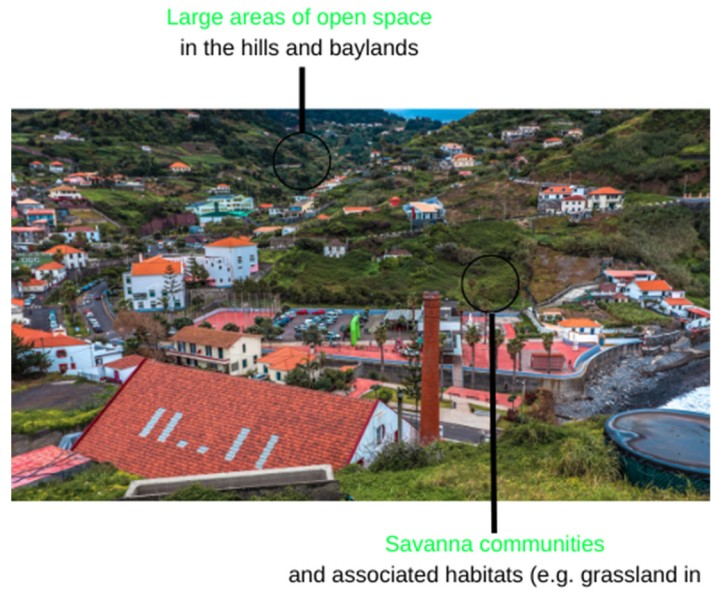

**Figure 17.** Parish of Porto da Cruz and its connection with nature.

Residents should be informed about the importance of landscape resilience and encouraged to participate in environmental conservation and ecosystem restoration initiatives. Additionally, it is essential that the local authorities responsible for implementing these measures work together with other local entities and organizations, including NGOs and community associations, to promote collaboration and dialogue among different stakeholders [17].

Promoting landscape resilience in Machico requires an integrated approach that includes conservation and restoration measures for ecosystems, adequate management of land use and water resources, community involvement, and implementation of sustainable agricultural practices. Some specific actions that can be taken include the restoration of degraded areas, the planting of native species, and proper waste management. It may also be important to promote awareness and environmental education among residents and visitors to the region to ensure the long-term sustainability of the landscape in Machico.

In the context of housing recovery actions, it is essential to recognize the interconnectedness between landscape resilience and the well-being of residents. The strategies discussed earlier, such as the creation of ecological protection zones, adoption of sustainable construction techniques, and promotion of environmental conservation practices, not only contribute to landscape resilience but also have direct implications for housing rehabilitation.

By emphasizing the integration of environmental considerations into housing strategies, we acknowledge that the restoration and enhancement of the landscape play a pivotal role in bolstering urban resilience. Moreover, these landscape-oriented measures can synergize with efforts aimed at housing recovery. For instance, the implementation of sustainable construction practices not only fortifies landscape resilience but also ensures the durability and habitability of dwellings, reducing the need for constant maintenance.

Furthermore, the promotion of environmental conservation practices and green infrastructure extends beyond aesthetic enhancement. These actions directly influence the thermal comfort of housing units, mitigating the energy consumption required for heating and cooling. This not only aligns with environmental sustainability but also reduces operational costs of homes, contributing to their accessibility and long-term viability.

Additionally, the reduction in disaster risks, as previously discussed, holds intrinsic connections with housing recovery. By constructing or renovating homes considering the specific natural risks of the region, we invest in the resilience of the structures themselves, thereby enhancing the safety and protection of residents.

Our comprehensive review meticulously addresses the synergy between landscape resilience and housing recovery actions, providing a comprehensive and cohesive understanding of the topic. This interconnectedness underscores the importance of an integrated approach to urban development, where landscape and housing initiatives complement each other, striving towards more resilient and sustainable communities.

The distinct characteristics of Machico's natural ecosystems warrant exploration. From the prominent Ponta de São Lourenço to the coastal expanses, dunes, and the unique geological formations, each ecosystem bears its own identity. We expound upon their significance, both ecologically and geologically, emphasizing their pivotal role in maintaining environmental equilibrium.

A vital discourse emerges around the intricate interaction between burgeoning urbanization and the existing natural ecosystems within Machico. We delve into how urban expansion can reverberate through these ecosystems, potentially leading to the loss of biodiversity, soil degradation, and disruptions in air and water quality. Amid urban planning and developmental strategies, we underscore the imperative of safeguarding these ecosystems.

Charting the course forward, we envision an approach that harmonizes specialized spatial analyses and cartography with Machico's local housing strategies (LHS). Recognizing the value of comprehending the interplay between the built and natural environments, we seek collaboration with spatial analysis and ecology experts. Their insights will enrich our study with detailed cartographic data and specialized spatial analyses, illuminating the complex dynamics at play.

Incorporating such an approach not only addresses the reviewer's recommendation for enriched cartography and spatial analyses but also exemplifies our unwavering commitment to refining our study's approach. By intertwining urbanization and natural ecosystems, our objective is to provide a holistic analysis of their interaction in the context of Machico. This comprehensive perspective aligns with our aspiration to contribute to the enhancement of urban and environmental resilience within the region. The amalgamation of these components stands as a testament to our commitment to robust scholarship and an all-encompassing understanding of the urban-nature interface in Machico. Figure 18 depicts a map featuring the identification of various eco sites in Machico.

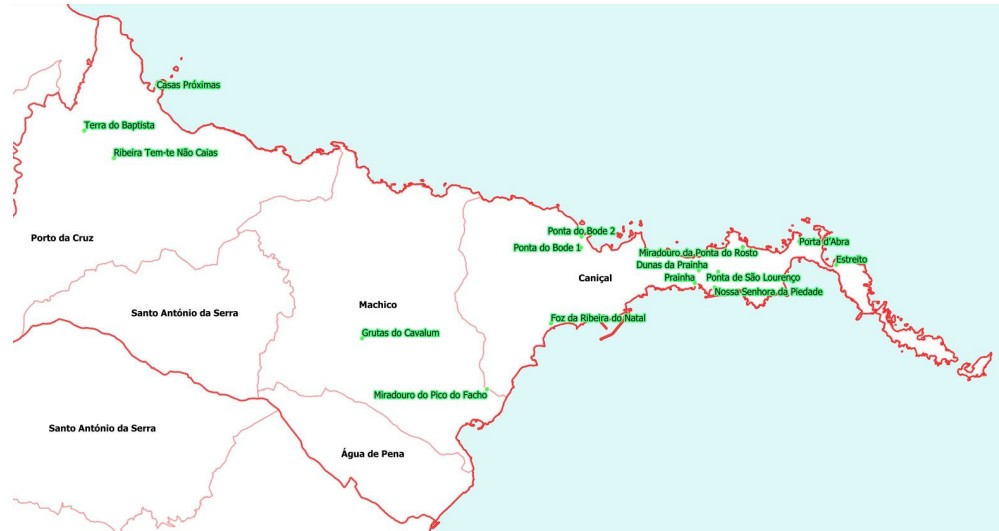

**Figure 18.** The eco sites in Machico (green).

Machico, situated in the beautiful Madeira archipelago, boasts diverse topography, ecosystems, and urban features that demand effective cartographic representation. By employing cartographic techniques, we aim to enhance the readers' comprehension of the intricate interactions between natural and built environments.

Accompanying this section is Figure 19, a detailed cartographic depiction of Machico. This map captures the geographic layout of the region, highlighting key landmarks, ecosites, urban areas, and natural features. The cartographic representation allows us to visualize the spatial proximity between housing clusters, natural reserves, coastlines, and other critical elements.

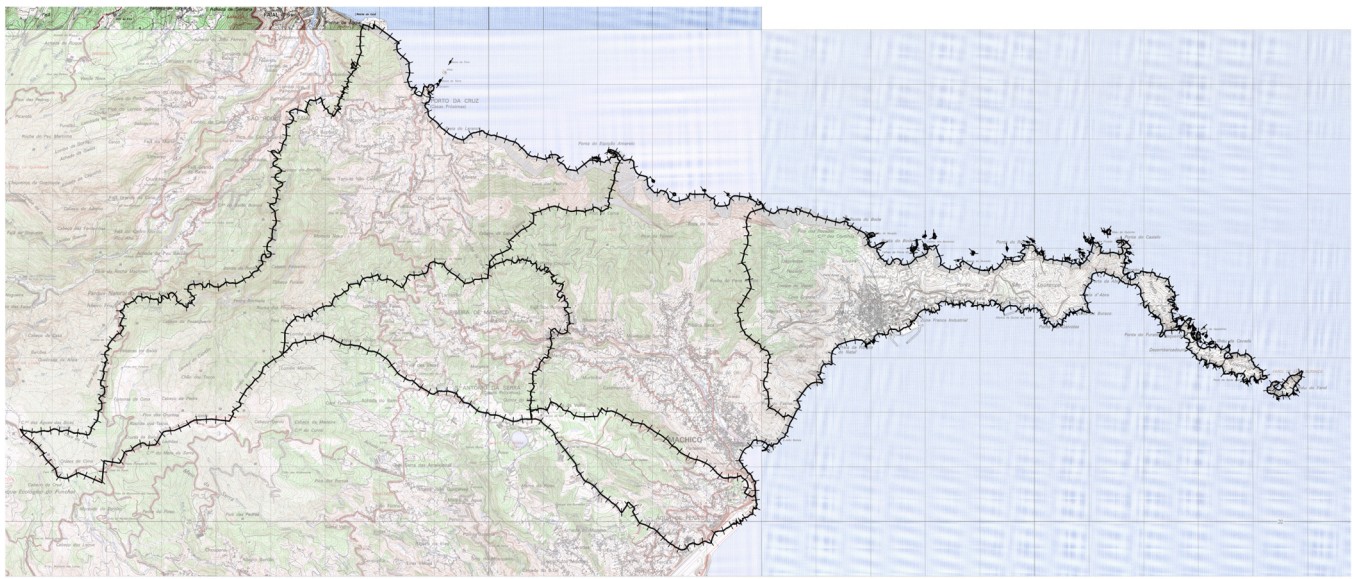

**Figure 19.** Cartography of Machico.

The cartography presented in Figure 19 introduces us to Machico through the lens of military cartography. This map encapsulates the city's spatial configuration, delineating its streets, squares, and prominent landmarks. As we explore this cartographic artifact, we uncover the historical progression of urban development, observing how the city's layout has transformed to accommodate evolving needs and trends.

The military maps provide a bridge between the past and the present, allowing us to compare the historical depiction of Machico with its contemporary state. By studying the changes and continuities reflected in these cartographic records, we gain insights into the city's growth trajectory, as well as the preservation of its cultural and natural heritage.

As we proceed with our study, the synergy between military cartography and our integrated approach becomes evident. These maps offer a tangible link between urbanization and natural ecosystems, helping us understand how the city's expansion has interacted with and, at times, transformed its surrounding landscape.

Incorporating these cartographic revelations into our exploration, we further enrich our understanding of Machico's narrative. By acknowledging the historical context provided by military maps, we strengthen our commitment to a holistic understanding of urbanization, ecology, and the dynamic interplay that characterizes this vibrant city.

The first map, Figure 20, provides an insightful overview of the diverse ecosites present within Machico. By delineating these ecologically significant areas, this map visually encapsulates the geographic distribution of natural ecosystems, allowing us to identify key hotspots of biodiversity and ecological importance.

Moving forward, Figure 21 showcases the protected areas within Machico. These designated regions play a critical role in conserving the region's natural heritage. The map offers a spatial understanding of the areas safeguarded for their unique ecological, geological, and cultural values, underscoring the importance of sustainable land management.

Lastly, Figure 22 delves into the geodiversity of Machico. By highlighting various geological formations and features, this map reveals the intricate geological history that has shaped the region's landscape over time. Understanding geodiversity contributes to a

holistic comprehension of the land's potential for various uses and its role in supporting ecosystem functions.

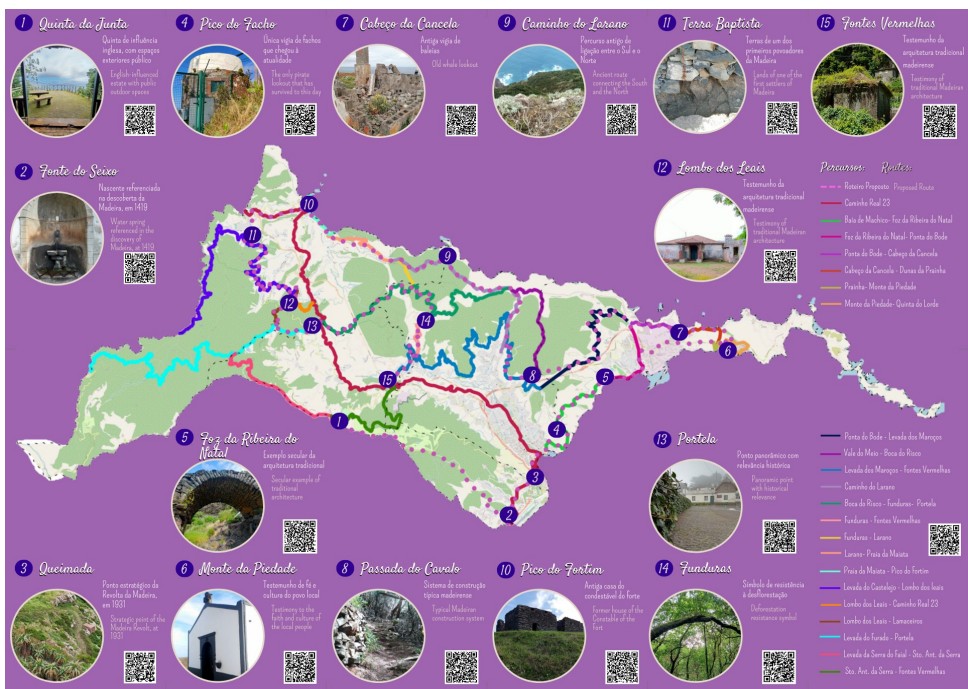

**Figure 20.** Ecosites Map [21].

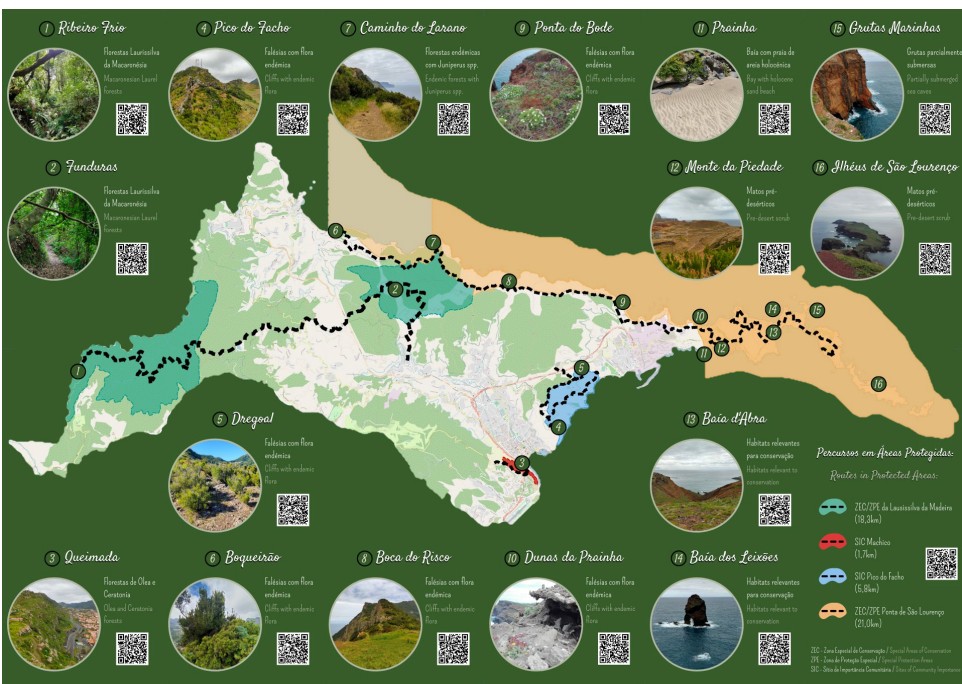

**Figure 21.** Protected Areas Map [22].

Through the integration of these cartographic representations, we endeavor to provide a comprehensive visual narrative of Machico's urban and natural landscapes. These maps collectively contribute to our exploration of the integrated approach between urbanization and natural ecosystems. While our current article may not extensively showcase these cartographic materials, the inclusion of these three maps underscores the importance of visual aids in enhancing our understanding of the intricate dynamics at play. As our study

progresses, we recognize the value of further incorporating robust cartographic analyses to deepen our insights into the complex relationships that define the region of Machico.

**Figure 22.** Geodiversity Map [23].

*2.3. Solutions for ELH in Conjunction with the Principles of the "1° Direito" Program*

According to the conducted diagnosis, it is necessary to develop housing solutions for 52 families, corresponding to 145 people; 39 of these families own homes that need rehabilitation due to unhealthiness, insecurity, and inadequacy conditions, while 13 are in rental and do not qualify for the "1° Direito" program. Table 11 shows that, out of these 39 owner families, 39 cases could be rehabilitated under the "1° Direito" program, as direct beneficiaries. The other 13 families will be supported through other rental market access programs, such as affordable rent. In addition to addressing the housing needs of the most vulnerable families, rehabilitation also aims to promote economic investment and population settlement, preserving the identity and heritage value of degraded urban spaces and old or historic urban spaces. The solutions also include support for families that need temporary relocation during interventions in their homes.

**Table 11.** Housing solutions to be promoted under the "1° Direito" [13].

| Framework | Occupancy Regime | Indignity Situation | Housing Solution | Promoter | Number of People |
|---|---|---|---|---|---|
| "1° Direito" | 39 Private | Unhealthiness and insecurity (37) | Rehabilitation | Direct Beneficiaries | 102 |
| | | Inadequacy (2) | | | |
| Rental Support Programs | 13 Renting | | | | 43 |

Table 12 aims to respond to specific challenges and goals. These goals are achieved through three main pillars: urban rehabilitation, housing rental, and housing qualification. These pillars guide public policy in the coming years and local housing strategies for the next 6 years.

**Table 12.** Pillars and Measures [13].

| | | |
|---|---|---|
| Urban Rehabilitation | Encourage lasting and regular conservation of buildings | Provide incentives for property owners to maintain and conserve buildings regularly. |
| | Reduce costs and simplify licensing for building rehabilitation | Streamline bureaucratic processes and reduce fees for building rehabilitation licenses. |
| | Attract investment to the rehabilitation of the housing stock | Promote tax incentives and financial support for housing rehabilitation projects. |
| Rental Housing | Boost the rental market | Implement initiatives to increase the availability and diversity of rental housing options. |
| | Integrate and valorize neighborhoods and social housing | Invest in the improvement of infrastructure, services, and public spaces in neighborhoods with social housing. |
| Santo António da Serra | Contribute to social inclusion and protection of the most disadvantaged | Develop affordable housing programs tailored to the needs of vulnerable populations. |
| | Address new social and demographic realities | Analyze and respond to evolving demographic trends and social challenges. |
| | Promote the improvement of housing conditions | Enhance the quality and safety of existing housing units. |

The new generation of housing policies (ENH) aligns with the objective of ensuring access to adequate housing for all while expanding the scope of beneficiaries and the size of the public-supported housing stock. The ENH emphasizes the primacy of building rehabilitation and urban revitalization as key intervention strategies. Various instruments are incorporated into the ENH, such as the "1º Direito" housing access support program, the Porta de Entrada program, the affordable rental program, Chave na Mão-housing mobility program for territorial cohesion, from housing to habitat, differentiated autonomous rates for long-term residential leases, and legislative changes to urban leases. Following a comprehensive diagnosis of housing needs, each challenge identified is addressed through these specific ENH instruments as outlined in the table.

According to Table 13, housing solutions were proposed to meet the identified needs, including the rehabilitation of family accommodations that can be recovered and adapted by the owners, and the mobilization of owners of vacant homes to rehabilitate and put them on the rental market at controlled costs, with tax benefits and security for tenants. A financial estimate was made for the rehabilitation of 39 homes for direct beneficiaries, with an estimated investment of 5.5 million euros.

**Table 13.** Financial estimate [13].

| | Gross Areas (by Typology) | Units | Total Gross Areas by Typology (m$^2$) | Rehabilitation € |
|---|---|---|---|---|
| Typology | m$^2$ | | | |
| T1 | 73 | 13 | 949 | 1,423,500.00 € |
| T2 | 95 | 13 | 1235 | 1,852,500.00 € |
| T3 | 117 | 12 | 1404 | 2,106,000.00 € |
| T4 | 128 | 1 | 128 | 192,000.00 € |

### 3. Results

This section is dedicated to the presentation and analysis of the findings and outcomes derived from our comprehensive investigation into the housing landscape of Machico. It is here that we unveil the culmination of our research efforts, shedding light on the current state of housing in the municipality, the identified deficiencies, and the strategic directions charted for the future.

In the preceding chapters, we embarked on a journey of diagnosis, dissecting the housing stock, supply and demand dynamics, access difficulties, and the multifaceted

intricacies of Machico's housing situation. We delved into the realms of SWOT analysis, outlining the municipality's strengths, weaknesses, opportunities, and threats, which serve as essential signposts for our strategic planning.

Moreover, our exploration extended to the concept of landscape resilience—a pivotal factor in ensuring the long-term sustainability and well-being of urban environments. We examined the symbiotic relationship between housing solutions and landscape resilience, recognizing their interdependence.

Additionally, we elucidated the strategic alignment between housing solutions and the "1° Direito" program, outlining financial frameworks and avenues for co-financing that facilitate access to dignified housing for those in need.

Now, as we transition into the Results section, we will articulate the tangible outcomes of our research endeavors. Here, we will provide a detailed account of the housing deficiencies identified in Machico, offering a comprehensive view of the challenges faced by the community. These findings serve as the bedrock upon which our future housing policies and programs will be constructed.

In tandem, we will outline our strategic blueprint for the future, presenting a vision of a housing landscape that is not only resilient but also socially inclusive and environmentally sustainable. These results encapsulate the essence of our commitment to ensuring that housing in Machico transcends the confines of shelter—it becomes a vehicle for improved quality of life, community well-being, and environmental stewardship.

### 3.1. Projecting Goals: A Desired Future

According to the municipal housing action plan of Machico, the goal is to achieve a desired future through the implementation of actions aimed at improving housing conditions for the population, revitalizing and rehabilitating degraded and historic urban spaces while preserving their identity and heritage value, boosting the rental market in the municipality, promoting inter-municipal mobility, consolidating occupancy in a manner compatible with existing infrastructure and buildings, and attracting and retaining working-age population. The action plan presented provides specific measures to achieve these objectives.

Table 14 provides an overview of the planned housing solutions to be implemented annually under the access to housing support program, or "1° Direito". These solutions include the construction of new buildings, the rehabilitation of existing buildings, the acquisition of housing for rental at controlled prices, and the mobilization of vacant properties. The table shows the number of planned housing solutions for each year, as well as the expected investment. This information is valuable for planning and ensuring the efficiency and effectiveness of housing actions.

**Table 14.** Forecast of implementation of housing solutions per year under 1° Direito [13].

| Solution | Housing Shortage | Occupation Regime | 2022 | 2023 | 2024 | 2025 |
|---|---|---|---|---|---|---|
| Rehabilitation of 39 private dwellings | Unhealthiness and insecurity (37) | Owners | T4: 1 | T1: 4 | T1: 4 | T1: 4 |
| | | | | T2: 4 | T2: 4 | T2: 4 |
| | | | | T3: 4 | T3: 4 | T3: 4 |
| | Inadequacy (2) | | T1: 1 | | | |
| | | | T2: 1 | | | |
| | Total | | 192,000 € | 1,962,000 € | 1,710,000 € | 1,710,000 € |

The urban landscape plays a crucial role in people's quality of life and the sustainable development of cities. According to Carmona [24], the urban landscape can be considered as the result of the interaction between the built and natural elements that make up the urban space and can directly influence the perception of inhabitants regarding the urban environment and its quality of life. In addition, the preservation and requalification of

historical and cultural heritage are considered important measures to promote the identity and cultural memory of cities.

Regarding the resilience of the urban landscape, the implementation of sustainable housing solutions can contribute to reducing the negative impacts of climate change and natural disasters, as well as promoting energy efficiency and reducing carbon emissions. The adoption of sustainable solutions for urban housing can contribute to the development of more sustainable and resilient cities, while promoting the quality of life of inhabitants.

Therefore, the consideration of the resilience of the urban landscape in Machico's local housing strategy can contribute to ensuring the long-term quality of life of inhabitants, while promoting sustainability and preservation of the urban environment. Monitoring the implementation of the ELH is essential to guide interventions, evaluate their effectiveness, and ensure compliance with the objectives set out in the schedule. The evaluation process included both an internal and external perspective. The executive team, as well as the social action, housing, urbanism, and territory planning teams, will be responsible for monitoring and internal evaluation. For external evaluation, we count on the support of IHRU (Housing and Urban Rehabilitation Institute) and ValeConsultores.

Continuous evaluation will include monitoring of the action plan, allowing for the evaluation of the achievement of the ELH objectives and making necessary adjustments. Post-intervention evaluation will measure the results and impacts generated by the interventions.

The involvement of all stakeholders, including technical teams, beneficiaries, public and private local entities, is essential to ensure commitments to action and change and to understand the potentialities and challenges in achieving the actions set out in the action plan. For this, continuous and ex-post monitoring, and evaluation of the implementation of the ELH will be carried out, using various processes, such as monthly meetings, semi-annual questionnaires, discussion groups, and quarterly reports. The evaluation will be carried out both internally, by the executive, social action, housing, and urbanism and territory planning teams, and externally, with the support of IHRU—Housing and Urban Rehabilitation Institute and consulting firm ValeConsultores. Ongoing evaluation will allow for the evaluation of the way ELH objectives are achieved and for any necessary adjustments or corrections, while post-intervention evaluation will measure the results and effects generated by interventions to keep housing problems resolved in the long term.

The relevance of the landscape resilience approach in territorial management and urban planning is highlighted by several authors. Landscape resilience involves the ability of the landscape to adapt and recover from disturbances and is an essential element to ensure the long-term sustainability of ELH interventions. The importance of monitoring landscape resilience and its impact on housing interventions is emphasized by Rey Benayas [25], who highlights the need to assess the landscape's ability to cope with disturbances and its regenerative capacity. To ensure the effective implementation of ELH, continuous evaluation and monitoring of landscape resilience and the impact of interventions on it are fundamental. Specific assessment tools and indicators are necessary to ensure the effectiveness of interventions and their ability to adapt to future changes. In this sense, the implementation of sustainable housing solutions that consider landscape resilience can ensure the durability and effectiveness of these interventions [26]. The integration of landscape resilience into the implementation of ELH can make this strategy more efficient, effective, and sustainable.

### 3.2. Achieving a Resilient Housing Future: Implementation and Assessment

In this section, we expound upon the goals of a desirable future characterized by resilient housing, as well as the systematic approach to implementing housing stock interventions. While we have outlined the planned analysis practices, we acknowledge the reviewer's astute observation regarding the necessity of elucidating the specific tools and ongoing assessment methods that will be employed in this process.

To address this imperative aspect, we introduce a dedicated subsection that outlines the methodologies and assessment tools pivotal to our proposed housing interventions. This serves to enhance the transparency and comprehensibility of our implementation strategy:

- Geospatial Analysis: We will employ geospatial analysis techniques to evaluate the existing environmental context, identifying areas of vulnerability and potential interventions. geographic information systems (GIS) will facilitate the spatial representation of critical factors such as land cover, topography, and natural hazards.
- Land-Use Modeling: Advanced land-use modeling will aid in simulating the potential impact of housing interventions on the local landscape and ecosystem. This will allow us to assess the feasibility and sustainability of proposed strategies prior to implementation.
- Community Engagement Methods: Meaningful community engagement will be fostered through surveys, workshops, and consultations with residents. These participatory approaches will ensure that the proposed housing strategies resonate with the needs and aspirations of the community while fostering a sense of ownership.
- Ongoing Assessment and Monitoring: Recognizing the dynamic nature of resilience-building efforts, we emphasize the significance of continuous assessment. We will establish a framework for ongoing monitoring, enabling us to track the effectiveness of implemented interventions, adapt strategies as necessary, and ensure alignment with landscape resilience goals.
- Quantitative and Qualitative Analysis: Our approach encompasses both quantitative metrics and qualitative insights. This dual-pronged analysis will enable a comprehensive evaluation of not only tangible outcomes but also the intangible sociocultural, economic, and ecological impacts of the interventions.

The implementation of housing stock interventions requires a rigorous assessment process to ensure their efficacy and alignment with landscape resilience goals. In this subsection, we outline our approach to continuous evaluation and highlight the significance of adapting to changing conditions.

- Adaptive Management Framework: Recognizing the complex and dynamic nature of urban environments, we will adopt an adaptive management framework. This framework acknowledges that unforeseen challenges and opportunities may arise during implementation. Regular reviews will allow us to modify strategies in response to evolving conditions.
- Indicators and Benchmarks: We will establish a set of indicators and benchmarks to measure the success of our interventions. These quantitative metrics will encompass factors such as improved green cover, enhanced water management, and increased community engagement. By comparing post-intervention data with baseline measurements, we can quantitatively demonstrate the impact of our actions.
- Longitudinal Studies: To capture the long-term effects of our housing interventions, we will conduct longitudinal studies that extend beyond the initial implementation phase. These studies will help us understand the lasting effects on landscape resilience, socioeconomic conditions, and community well-being.
- Collaborative Partnerships: Collaboration with local research institutions, environmental experts, and community stakeholders will enhance the robustness of our assessment efforts. Engaging diverse perspectives will contribute to a holistic evaluation of the interventions' outcomes.
- Knowledge Sharing and Transparency: We are committed to transparency in sharing our assessment findings with the community and relevant stakeholders. Regular updates and reports will provide insight into the progress and outcomes of our initiatives, fostering accountability and building trust.

## 4. Discussion

This study embarked on an in-depth exploration of housing deficiencies and landscape resilience in the Municipality of Machico, unveiling a comprehensive overview of the

current state of housing, challenges faced by the community, and strategic directions for the future. The discussion section delves into the key findings and their implications, underscoring the significance of addressing housing issues within a broader framework of sustainability, resilience, and community well-being.

- Housing Deficiencies and Access Difficulties: The diagnosis of housing deficiencies and access difficulties revealed a multifaceted landscape in Machico. The characterization of the housing stock illuminated the presence of unhealthiness, insecurity, and inadequacy in various dwellings. It is evident that a substantial portion of the population resides in conditions that lack dignity, underscoring the urgency of intervention. The supply and demand dynamics highlighted the need for a multi-pronged approach that encompasses building rehabilitation, property renting, construction, and property acquisition. The discrepancies between supply and demand emphasize the importance of flexible and diverse housing solutions that cater to the unique needs of the community.
- Landscape Resilience as a Cornerstone of Housing Solutions: The integration of landscape resilience into housing solutions emerged as a central theme in this study. The discussion underscores that the landscape is not a passive backdrop but a dynamic system that significantly influences the quality of life and well-being of residents. Promoting landscape resilience can mitigate the impacts of extreme events, enhance environmental sustainability, and foster a stronger connection between residents and their natural surroundings. The study advocates for the consideration of landscape resilience as an integral element of adequate housing. Measures such as the rehabilitation of existing buildings with environmental sustainability solutions and the creation of green spaces can contribute to risk mitigation and a healthier urban environment. Furthermore, the assessment of landscape risks is essential for informed decision-making and long-term problem prevention.
- The "1° Direito" Program and Financing Conditions: The discussion on the "1° Direito" program highlights its commendable objective of ensuring access to suitable housing for individuals facing financial constraints and residing in undignified conditions. The program offers a range of strategies, from building rehabilitation to property acquisition, to address the diverse needs of the community. The financing conditions outlined in Table 15 play a pivotal role in facilitating access to dignified housing. These conditions, with non-repayable co-financing and investment by project promoters, are meticulously designed to address specific criteria and eligibility standards. Notably, the inclusion of accessibility and environmental sustainability provisions aligns with the broader goal of promoting sustainable and resilient housing solutions.
- Towards a Resilient and Sustainable Future: In the pursuit of a resilient and sustainable housing future, this study underscores the importance of a holistic approach. Housing solutions should not be viewed in isolation but as integral components of a broader urban landscape. By considering landscape resilience, we can create housing environments that not only provide shelter but also enhance the well-being of residents and contribute to environmental sustainability. The collaborative efforts of various stakeholders, including local authorities, environmental experts, and community members, are essential in realizing this vision. Engaging diverse perspectives fosters a sense of shared responsibility and empowers communities to actively shape their living environments while safeguarding natural surroundings.

**Table 15.** Financing conditions under the "1° Direito" program [13].

| Solution | Non-Repayable Co-Financing | Total Investment by Promoter Using a Loan with a 50% Interest Rate Subsidy | Schedule |
|---|---|---|---|
| Rehabilitation | 40% | 60% | |
| Construction | 35% | 65% | |

| Solution | Non-Repayable Co-Financing | Total Investment by Promoter Using a Loan with a 50% Interest Rate Subsidy | Schedule |
|---|---|---|---|
| Acquisition of buildings | 30% | 70% | Until 2026 |
| Acquisition of land | 35% | 65% | |
| Rental | 50% for the first 5 years / 25% between 5 and 10 following years | Not Applicable | Up to 10 years after approval of the application to the "1º Direito" program |

### 4.1. Framing of the Measures of the "1º Direito"

The housing access support program, known as 1º Direito, aims to ensure access to adequate housing for people living in undignified conditions and who do not have the financial capacity to purchase a home without assistance. To achieve this goal, the program provides various solutions, including the rehabilitation of existing buildings, rental, construction, and acquisition of properties. These solutions can be carried out by both public entities and households in a self-promotion regime. To be eligible for financing under the program, these solutions must meet certain criteria, such as the legalization of constructions and the demolition of unlicensed buildings. In addition, eligible expenses include the price of acquisitions or contracts, work and supplies necessary for accessibility and environmental sustainability solutions, services related to projects, supervision, and safety of the work, and notarial and registration acts.

Landscape is seen as a complex system capable of adapting and resisting natural or human-induced disturbances, with its resilience being a crucial factor for the safety and well-being of communities [27]. Promoting landscape resilience can help mitigate the effects of extreme events, such as floods or landslides, providing a safe and healthy environment for people [28]. In addition, the landscape can have a positive impact on the mental and physical health of residents, promoting their connection with nature and the creation of green spaces [29].

In the context of the 1º Direito program, it is important to consider the landscape as an integral element of adequate housing and the quality of life of communities. The implementation of measures that promote landscape resilience can contribute to risk mitigation and the creation of a healthier and more sustainable environment. This can be achieved through the rehabilitation of existing buildings with the implementation of environmental sustainability solutions, the creation of green spaces, and the adoption of measures to adapt to extreme events [30]. The assessment of the landscape and its risks is also essential for taking appropriate measures and preventing future problems [27].

Therefore, promoting landscape resilience should be a central concern within the scope of the 1º Direito program, contributing not only to the quality of life of residents but also to the sustainability of the environment and the resilience of communities in crisis situations.

### 4.2. Lines of Financing and Co-Financing for ELH in the Municipality of Machico

Table 15 within the framework of the "1º Direito" program encompasses financing conditions for housing solutions to facilitate access to suitable housing for individuals living in inadequate conditions, and it involves co-financing from national administrations only and plays a pivotal role in delineating the financial provisions that underpin housing solutions. These solutions are meticulously designed to ensure access to suitable housing for individuals residing in conditions that lack dignity and who face financial constraints hindering their ability to secure appropriate housing without external support. The program offers a comprehensive spectrum of strategies, encompassing the realms of building rehabilitation, property renting, construction endeavors, and property acquisition.

Elaborating on the specifics, Table 15 elucidates the financing conditions governed by the "1º Direito" program [13]. These conditions, in a collaborative effort between the central

administration and stakeholders, outline the nuances of non-repayable co-financing and the comprehensive investment responsibility borne by project promoters. Such investment is often facilitated through loans subsidized with a 50% interest rate reduction.

Distinct housing strategies are meticulously allocated within these conditions:

- Rehabilitation, which involves 40% non-repayable co-financing and a 60% investment by promoters until 2026;
- Construction, with 35% non-repayable co-financing and 65% investment by promoters;
- Acquisition of buildings, requiring 30% non-repayable co-financing and 70% investment by promoters;
- Acquisition of land, with 35% non-repayable co-financing and 65% investment by promoters.

For rental housing solutions, a unique approach applies. Over the first five years, a substantial 50% non-repayable co-financing is provided, followed by 25% co-financing for the subsequent five years. It is important to note that these financing conditions cater to specific requirements outlined in Table 15, encompassing criteria such as the acquisition of land and housing properties to ensure suitable housing in alternate locations. Additionally, these conditions encompass provisions for accessibility and environmental sustainability solutions. Eligible expenses encompass various aspects, including acquisitions, contractual agreements, essential work and supplies, project-related services, construction supervision, safety measures, as well as notary and registration acts.

Furthermore, Table 15 delineates eligibility criteria for financing, encompassing adherence to existing regulations and standards, along with the requirement of approval by the competent municipality. This table stands as a crucial tool, instrumental in guaranteeing access to dignified housing for those most in need, thereby fostering an environment that is secure and conducive to well-being.

The imperative of landscape resilience cannot be overstated in safeguarding urban inhabitants' safety and welfare, while concurrently mitigating the threats posed by natural disasters. Research underscores that landscape resilience presents a promising paradigm for adapting and restoring areas impacted by extreme events, while simultaneously supporting biodiversity preservation and ecosystem services [31]. In the evaluation of housing projects, the incorporation of landscape resilience emerges as a pivotal criterion. This approach, encompassing the creation of more resilient and sustainable housing solutions, significantly diminishes the vulnerability of urban areas to extreme events, thereby fortifying community resilience [32,33].

Moreover, practices aimed at conserving and restoring natural ecosystems, including the establishment of green spaces and afforestation initiatives, present an effective means to enhance landscape resilience and reduce urban vulnerability. The inclusion of landscape resilience as an evaluative criterion within the "1° Direito" program can undeniably contribute to fostering sustainable and resilient housing solutions within urban landscapes. Such an integration ensures an environment that is both secure and conducive to the long-term well-being of communities.

## 5. Conclusions

Participation in the rehabilitation, construction, or acquisition of properties is exclusively for housing areas, as defined in Article 4.8 of Decree-Law N° 37/2018 [34]. It is intended for owner-occupied and permanent housing, supported leasing, conditional rent, reduced rent through special programs, or resoluble ownership. In the case of residential units, eligible expenses include the entire area of the building or fraction intended for housing. Expenses for accessibility and environmental sustainability are also considered for the calculation of the participation. Rehabilitation interventions are only eligible for funding if they allow for an increase in the energy class by at least two levels. Additional increases are only applied exceptionally, such as in the case of subleasing.

Participation can be requested by owners, tenants, building administrators, or other legal entities, provided that the conditions established by Decree-Law N° 37/2018 [34]

are met. It is important to note that participation requests must be submitted before the start of works, and that the amount of participation will be paid after their completion and inspection.

In addition to the conditions established by Decree-Law Nº 37/2018 [34], other additional conditions may also apply depending on the specific characteristics of each project or intervention. For example, in the case of new building construction, it may be required that they comply with certain energy efficiency and accessibility standards. Participation can also be limited to a certain number of projects per year or per region. It is important that interested parties carefully verify all conditions and requirements before submitting a participation request.

Regarding the importance of urban and natural landscape resilience, Davoudi [35] highlights that resilience is an essential approach for dealing with the uncertainty and risks associated with environmental and social change in urban areas. Furthermore, landscape resilience has increasingly been recognized as an effective strategy for addressing global environmental challenges such as climate change [28]. Landscape resilience is defined by several dimensions, such as biodiversity, connectivity, and water absorption capacity [16]. These dimensions are essential for maintaining ecosystem integrity and providing ecosystem services to urban communities.

To enhance landscape resilience, measures such as the use of sustainable construction techniques and the planting of trees and green areas are recommended in the scientific literature. A study conducted by Escobedo [36] suggests that the preservation and planting of trees can help reduce urban temperature, improve air quality, and increase urban landscape connectivity. However, it is essential that construction and rehabilitation policies include measures to enhance landscape resilience. This can not only improve the quality of life of people who depend on the landscape but also help protect the natural ecosystems that sustain life on Earth.

The first recommendation for future housing policies and programs is the implementation of social inclusion measures, which includes ensuring access to affordable and safe housing for all layers of society, regardless of their income or socioeconomic status. This can be achieved through the creation of housing subsidy programs, construction of popular housing, and implementation of social rent measures.

The second recommendation is the promotion of environmental sustainability measures, which covers the construction of environmentally friendly housing, such as low-energy consumption buildings and renewable energy sources, as well as prioritizing construction in already consolidated urban areas to avoid the development of new urban areas and environmental degradation.

A third recommendation for shaping future housing policies and programs centers around advocating for landscape resilience in urban areas. Landscape resilience encompasses the capacity of urban regions to withstand the impacts of climate and the environment, including challenges like floods, droughts, and landslides. Achieving this resilience involves adopting nature-based solutions, including the establishment of green spaces and the promotion of urban biodiversity. Additionally, incorporating sustainable and energy-efficient construction techniques contributes to this effort. The proposal to enhance landscape resilience presents substantial potential for its seamless integration into home renovation projects, thereby bolstering both the resilience of urban settings and nurturing sustainability.

The collaborative efforts of a diverse spectrum of stakeholders play a pivotal role in realizing this integration. Local municipalities and authorities, environmental agencies, and non-governmental organizations (NGOs), financial institutions, private sector collaborations, government initiatives and incentives, as well as community and resident endeavors—all these entities are instrumental in financing and executing these interventions. The harmonization of landscape resilience with home renovation initiatives aligns seamlessly with the overarching aspiration of cultivating urban areas that are both more

resilient and sustainable. This endeavor requires the engagement of a varied assembly of stakeholders and the cultivation of cross-sector cooperation.

In this cooperative approach, communities ensure that renovation undertakings not only enhance individual residences but also collectively contribute to the establishment of a more resilient urban fabric. This collaborative stance fosters a shared sense of responsibility, empowering both individuals and communities to actively shape the trajectory of their living environments while safeguarding the natural surroundings.

Finally, it is important that housing policies and programs are developed collaboratively and participatively, involving local and community actors to ensure that local needs and demands are effectively met. This includes creating community housing committees, consulting with residents and local organizations, and ensuring transparency and popular participation in housing-related decisions. Landscape resilience can also be achieved through the promotion of resilient communities that are able to quickly adapt and recover from extreme events such as floods that occur throughout the island, not only in Machico [14], but also in Ribeira Brava [37], and São Vicente [38]. This includes strengthening community ties, promoting economic diversity, and creating social safety nets. Additionally, it is important that housing policies and programs are developed considering the specificities of each urban area, to ensure that solutions are tailored to local needs and effective in promoting landscape resilience.

**Author Contributions:** Conceptualization, R.A. and S.L.; methodology, R.A.; validation, S.L., J.C. and J.M.N.G.; formal analysis, S.L.; investigation, R.A.; resources, R.A.; writing—original draft preparation, R.A.; writing—review and editing, R.A.; visualization, R.A.; supervision, S.L.; project administration, S.L.; funding acquisition, R.A. All authors have read and agreed to the published version of the manuscript.

**Funding:** This research was created within the framework of the project 2021-1-CZ01-KA220-HED-000031187 "ESDGs! Sustainable Development Goals in Education in Action!".

**Data Availability Statement:** We are committed to promoting transparency and reproducibility in research. The data supporting the results reported in this article are available in accordance with MDPI's data sharing policies. The datasets generated and/or analyzed during the study are publicly archived and can be accessed through the following link: [https://www.cm-machico.pt/]. If no new data were created or if data availability is restricted due to privacy or ethical considerations, we affirm that this statement is still required for transparency. We have provided detailed information on how to access or obtain the data whenever possible, and we are willing to provide additional information upon request to ensure the reproducibility of our research.

**Acknowledgments:** We want to thank the project "ESDGs! Sustainable Development Goals in Education in Action!" (2021-1-CZ01-KA220-HED-000031187) for providing the essential financial and logistical support for the completion of this research. Without the funding and resources made available by this project, this study would not have been possible.

**Conflicts of Interest:** The authors declare no conflict of interest.

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
