# Peer review of "Local Housing Strategy: Analysis of Importance and Implementation in Machico Municipality, Madeira"

_land, doi:10.3390/land12091778_

Round 1

Reviewer 1 Report

Overall, I am unclear what you are trying to achieve with this paper.  It includes incredibly granular analysis of the housing situation in Machico, but does not provide a framework that uses that data to make a point.  You need to think through what exactly the argument you are making is and include enough detail and organization that the reader can follow your argument without getting lost in the minutiae.

.

Some specific items to address are below:

116 Reference to T3 typology with no explanation of system

174 RGEU4 (again no explanation)

 Supply and Demand section needs reorganization.  Its not clear how it is organized and some sections, such as the paragraph on tourism does not seem to address housing supply or demand 242-247.  Elsewhere there are many specific facts, but they have not been organized into a clear argument about housing in Machico

 Unclear relationship between housing focus and landscape restoration.

Table 13, are those amount really euro per square metre?

Some strange language choices (perhaps an artefact of translation).  For example, 264 “average price practiced”

Author Response

Thank you for your observations and for the opportunity to improve our manuscript!
We are very grateful for taking the time to analyze the paper and make very useful, encouraging, and thoughtful comments and recommendations.
We have read the evaluation carefully and, based on the review report, we performed revisions of our manuscript, as requested, the modifications being highlighted with green into the manuscript and in the answers bellow.

Reviewer 2 Report

The manuscript deals with an interesting topic

Some integrations are requested from the authors

1.        Introduction: it would be useful to report, in the introduction or in a separate paragraph, which are the policies at national level that aim to improve housing conditions

2.       specify the types indicated with T1, T2, etc. in figure 1; the classification is not clear

3.       line 127: it would be useful if the classification scale of the status of single-family housing were specified; for example, which conditions correspond to a "poor" condition?

4.       Line 174: what does RGEU4 mean?

5.       Line 199: what does Q1 and Q4 (before the year) mean?

6.       Table 4: explain better hoe the affordability index is computed. For example, when the household is composed by a couple, the median wage is considered double?

7.       Table 7: specify the meaning of “Isolated families”, “nuclear families”, etc.

8.       Line 367: specify the acronym “IAS”

9.       Paragraph 3 deals with the resilience of the landscape, with some examples; however, its relationship with the actions described below relating to the recovery of homes is not clear.

10.   Table 8: specify better the data (for example: how is computed the Max rent?)

11.   Par. 2.4 describe better the SWOT Analysis and the results obtained (strengths, weaknesses, etc.).  who participated in the analysis? (citizens, public administrators, local real estate agencies  ….?)

12.   Par. 2.1 How the PDM and PERU Programs considers the landscape resilience?

13.   Table 12: reports the pillars and the measurements. It will be useful report also the actions that that make it possible to achieve the goals (for example new social and demographic realities).

14.   Table 13: separate the unit cost from the total costs

15.   Table 15: the economic resources come from local or national administration? 

16.       In conclusion, the recommendation for the promotion of landscape resilience is reported. How can they be integrated with home renovation projects? (for example: who are the subjects who can finance and carry out this type of intervention?

Author Response

Thank you for your observations and for the opportunity to improve our manuscript!
We are very grateful for taking the time to analyze the paper and make very useful, encouraging and thoughtful comments and recommendations.
We have read the evaluation carefully and, based on the review report, we performed revisions of our manuscript, as requested, the modifications being highlighted with blue into the manuscript and in the answers bellow.

Reviewer 3 Report

The article analyzes the impact of the implementation of "Local Housing Strategies" in Machico, Madeira. It is a very interesting and valuable work, however, it lacks scientific rigor.
The paper explains the objectives of "Local Housing Strategies", but does not detail the objective of the research work, nor does it adequately explain the methodology used. The paper is of great interest from a heritage and architectural point of view, however, it is not a scientific article but a case study.
Due to the high interest of the work, the authors are invited to restructure the document, providing it with scientific methodology and resubmit it for publication.

Minor editing of English language required

Author Response

Thank you for your observations and for the opportunity to improve our manuscript!
We are very grateful for taking the time to analyze the paper and make very useful, encouraging and thoughtful comments and recommendations.
We have read the evaluation carefully and, based on the review report, we performed revisions of our manuscript, as requested, the modifications being highlighted with yellow into the manuscript and in the answers bellow.

Reviewer 4 Report

The paper provides a descriptive and prescriptive overview of an urban housing recovery policy program ivolved in the preservation of landscape resilience in the municipality of Machico in Madera.

The paper is not organized according to the usual pattern: introduction, materials, method, application, results, discussion, and conclusion. The first part is relevant as to the proper analysis of the built environment and the description of the overall state of the building fabric in the different neighborhoods and promises the testing of an approach that on the basis of these results, through a robust evaluation, both quantitative and qualitative, can provide decision-making and planning tools to the administration for the application of the program with a view to Landscape Resilience. Consistently, the authors provide Priority Lines of intervention that inform municipal policy.

Below, the authors list some of the critical spatial environmental issues by which they intend to represent the aspects of landscape fragility that housing policies address.

Below the authors list some of the critical spatial environmental issues by which they intend to represent the aspects of landscape fragility that housing policies address.

In Actions 3 and 4, the authors set out in a very general way what landscape resilience solutions need to be adopted for the context studied but do not present studies of any nature and cartographic, numerical representations and references to the actual state of the environmental spatial support to which the local landscape resilience relates and thus can be somehow measured and represented in view of the suggested practices.

The photos provided are insufficient to explain, represent, and quantify (as done with tables and graphs for the social, real estate, and housing contexts) the vulnerability of the land for which the program intends to remedy.

In section 5, the authors indicate the goals of a desirable future, with reference to the implementation program of housing stock interventions by indicating the planned analysis practices but not the tools, especially the ongoing assessment tools.

Overall, neither the research question nor the scientific contribution of the authors are clear. They very effectively raise the housing and spatial issue in the initial part of the paper, but only describe local policies and provide general recommendations on the need to preserve landscape resilience through participation as well.

Also for the purposes of the special issue, the paper should at least be supported by cartographic materials and spatial analysis specialized in the field of natural ecosystems with which the urban system dialogues.

Author Response

Thank you for your observations and for the opportunity to improve our manuscript!
We are very grateful for taking the time to analyze the paper and make very useful, encouraging and thoughtful comments and recommendations.
We have read the evaluation carefully and, based on the review report, we performed revisions of our manuscript, as requested, the modifications being highlighted with gray into the manuscript and in the answers bellow.

Round 2

Reviewer 1 Report

I appreciate the changes that you have made to the language and clarity of the paper.  The terms are much clearer and the flow in certain areas is much improved.  However, I have trouble determining the central argument of your paper.  You have done an extensive analysis of the lack of housing quantity and quality in Machico and put it in the context of existing policy, but what are your specific results? What theory/ideas are you testing?  How does this expand the reader's understanding of housing policy and its effects?  

Author Response

Thank you for your observations and for the opportunity to improve our manuscript!
We are very grateful for taking the time to analyze the paper and make very useful, encouraging, and
thoughtful comments and recommendations.

Reviewer 2 Report

11.       Introduction: it would be useful to report, in the introduction or in a separate paragraph, which are the policies at national level that aim to improve housing conditions

AUTHORS: OK

2.       specify the types indicated with T1, T2, etc. in figure 1; the classification is not clear

AUTHORS: OK

3.       line 127: it would be useful if the classification scale of the status of single-family housing were specified; for example, which conditions correspond to a "poor" condition?

AUTHORS: OK

4.       Line 174: what does RGEU4 mean?

AUTHORS: OK

5.       Line 199: what does Q1 and Q4 (before the year) mean?

AUTHORS: OK

6.       Table 4: explain better hoe the affordability index is computed. For example, when the household is composed by a couple, the median wage is considered double?

AUTHORS: OK

7.       Table 7: specify the meaning of “Isolated families”, “nuclear families”, etc.

AUTHORS: OK

8.       Line 367: specify the acronym “IAS”

AUTHORS: OK

9.       Paragraph 3 deals with the resilience of the landscape, with some examples; however, its relationship with the actions described below relating to the recovery of homes is not clear.

AUTHORS: OK

10.   Table 8: specify better the data (for example: how is computed the Max rent?)

AUTHORS: OK

11.   Par. 2.4 describe better the SWOT Analysis and the results obtained (strengths, weaknesses, etc.).  who participated in the analysis? (citizens, public administrators, local real estate agencies  ….?)

AUTHORS: OK

12.   Par. 2.1 How the PDM and PERU Programs considers the landscape resilience?

AUTHORS: OK

13.   Table 12: reports the pillars and the measurements. It will be useful report also the actions that that make it possible to achieve the goals (for example new social and demographic realities).

AUTHORS: OK

14.   Table 13: separate the unit cost from the total costs

AUTHORS: OK

15.   Table 15: the economic resources come from local or national administration?

AUTHORS: OK

16.   In conclusion, the recommendation for the promotion of landscape resilience is reported. How can they be integrated with home renovation projects? (for example: who are the subjects who can finance and carry out this type of intervention?

AUTHORS: OK

Author Response

(The authors gave the same response as above.)

Reviewer 3 Report

The authors have made an effort to give the manuscript a more scientific structure. If the manuscript is approved by the editors, it is suitable for publication.

Author Response

(The authors gave the same response as above.)

Reviewer 4 Report

The authors did their best to improve the scientific soundness of the paper listing the tools that in future can be applied. They provided some more graphic materials 

Author Response

(The authors gave the same response as above.)
